# Thinking with Visual Abstract: Enhancing Multimodal Reasoning via Visual Abstraction

## Abstract

Images usually convey richer detail than text, but often include redundant information, which potentially downgrades multimodal reasoning performance. When faced with lengthy or complex messages, humans tend to employ abstract thinking to convert them into simple and concise abstracts. Inspired by this cognitive strategy, we introduce a novel paradigm to elicit the ability to Think with Visual Abstract (VAT), by prompting Multimodal Large Language Models (MLLMs) with visual abstract instead of explicit verbal thoughts or elaborate guidance, permitting a more efficient visual reasoning mechanism via concentrated perception. VAT encourages models to focus on more essential visual elements, concepts and structural features by undermining redundant information compared with explicit thinking methods, such as Chain-of-thought (CoT) and tool-using approaches, that increase the complexity of reasoning process via inserting verbose intermediate steps and external knowledge. Experimental results show that VAT consistently empowers different MLLMs in visual perception and reasoning tasks. VAT achieves an average gain of 2.21% over GPT-5 baseline, surpassing the gain of CoT, demonstrating that VAT better enhances multimodal task performance of MLLMs. Additionally, VAT spends fewer tokens while achieving higher performance. These findings highlight the effectiveness of visual abstract thinking and encourage further exploration of more diverse reasoning paradigms from the perspective of human cognition.

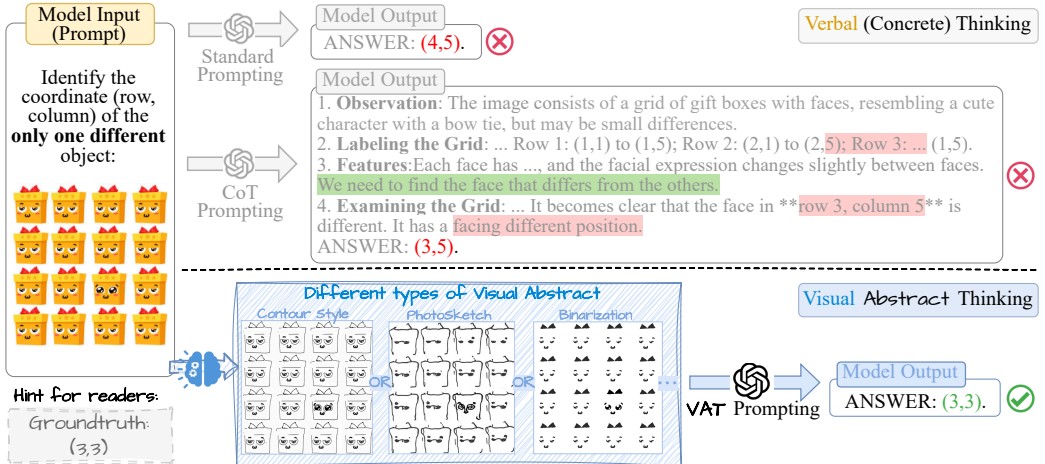

Figure 1: Illustration of VAT compared with standard and CoT prompting that elicit verbal thinking. VAT provides MLLMs with additional abstract perceptual context to encourage models to think through *visual abstract* that retains essential visual elements, and enables efficient reasoning for the correct answer without generating verbose intermediate steps.

## 1 Introduction

Recent advances in Multimodal Large Language Models (MLLMs) (Bai et al., 2025; OpenAI, 2025; Comanici et al., 2025) demonstrate impressive competence across a wide range of multimodal tasks, including perception (Fu et al., 2024; Wang et al., 2025b), grounding (Li et al., 2025b; Chen et al., 2023), and reasoning (Ma et al., 2025; Guan et al., 2024; Zhu et al., 2025). To further improve

multimodal reasoning performance, many existing approaches supply inputs with concrete and detailed verbal or visual guidance, verbalize concrete intermediate rationales, or provide additional visual information through In-Context Learning (ICL) (Brown et al., 2020; Dong et al., 2024), Chain-of-thought (CoT) (Wei et al., 2022; Lightman et al., 2024), and their multimodal variants (Wang et al., 2023; Zhang et al., 2024; Zhou et al., 2024). Tool-using methods are also widely investigated to further enhance multimodal perception and reasoning via tools, adding their corresponding textual or visual outputs into the context (Hu et al., 2024a; Zhou et al., 2024; Yang et al., 2023b).

Unlike textual modality, visual inputs such as raw images naturally contain richer information and far more details than required by a single reasoning problem. In some cases, this richness of information can lead to excessive details that overwhelm models and introduce distracting redundancy, impeding effective reasoning (Ling et al., 2023). The aforementioned CoT-based and tool-using methods can be concluded as ***explicit thinking***, as these methods rely on plenty of intermediate verbal steps and supply models with external and additional information. Although explicit thinking paradigms can improve task performance and interpretability (Wei Jie et al., 2024; Lu et al., 2024), the detailed verbal steps and external visual context further enlarge the involved volume of information, unintentionally amplifying redundancy, and potentially decreasing the efficiency (Yue et al., 2024; Xu et al., 2024).

To complement explicit thinking, humans faced with visually complex or lengthy inputs develop an opposite manner, ***abstract thinking***, to perceptually compact massy information into simpler or conceptual forms (Susac et al., 2014; Breuning, 2003; Tversky, 2013). Perception forms the foundation of multimodal understanding Bisk et al. (2020). Enhancing perception not only leads to more accurate recognition but also supports stronger downstream reasoning, making perception and reasoning more tightly coupled in multimodal contexts (Marr, 2010). Drawn on this cognitive strategy, Figure 1 demonstrates that abstract thinking from the visual side also empowers multimodal reasoning in MLLMs. In this example, the target image (gift boxes) is represented as either of the visual abstracts, retaining essential features while omitting surface details. Some of them appear completely different from the original image, but they correctly depict the contours, facial and relational information, and eventually guide the model to the correct answer. We name this reasoning mode as ***thinking with visual abstract***, as it helps focus cognition on essential visual elements towards more effective and efficient multimodal reasoning.

In this paper, we propose VAT, a novel paradigm for MLLMs to replace verbose elaboration with simple visual abstract during the reasoning process. In detail, we transform input images into visual abstracts via pixel-based conversions (Canny, 1986) or semantic models (Chan et al., 2022; Li et al., 2019), and integrate these visual abstracts in a simple prompt requiring models to generate responses without step-by-step slow thinking. VAT thus serves as a visual counterpart to CoT, emphasizing the potential of visual abstracts over accumulations. Through VAT, we investigate whether current MLLMs elicit similar ability to think in an abstract way as humans do. Specifically, we discuss: (1) how visual abstracts assist in multimodal tasks, (2) the impact of *visual abstract thinking* across tasks and models, and (3) the flexibility and efficiency of VAT. Our contributions are as follows:

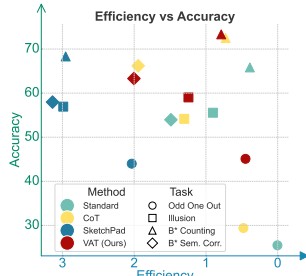

Figure 2: Time and budget costs of VAT compared to standard prompting, CoT, and the tool-using method Visual SketchPad (SketchPad), using GPT-4o. B*: BLINK.

- We introduce **VAT**, a new paradigm designed to align visual perception of MLLMs with human-like *visual abstract thinking* strategy. This encourages MLLMs to reason with abstract visual perception, which consequently enhances multimodal perception and reasoning performance.
- Our **VAT** achieves an average accuracy gain of +2.21 over GPT-5 baseline, compared with +0.96 of CoT. VAT is especially effective in tasks involving object relation and spatial reasoning, by highlighting contour and structural information.
- **VAT** is both time- and cost-effective as shown in Figure 2. On GPT-4o, it requires only ×0.28 and ×0.68 runtime of Visual Sketchpad and CoT, respectively, while consuming just ×0.1 tokens compared with Visual Sketchpad.

## 2 RELATED WORKS

**Chain-of-Thought and Multimodal Extensions.** Chain-of-Thought (CoT) prompting (Wei et al., 2022) encourages LLMs to produce rationales and progressive steps before arriving at a final answer.

Following works extend CoT into more diverse tasks and scenarios (Yao et al., 2023a; Besta et al., 2024; Ranaldi et al., 2024; Wang et al., 2024), including multimodal domain (Zhang et al., 2024; Li et al., 2025a). These methods incorporate natural language explanations into the reasoning trajectory (Ding et al., 2024; Xu et al., 2025; Aytes et al., 2025) to tackle complex reasoning tasks. While multimodal variants augment the reasoning chain with external images or dense region descriptions (Yang et al., 2023a; Wu et al., 2024; Shtedritski et al., 2023), more recent work has proposed to thinking with images. For example, they employ models to produce their own visual intermediates (Li et al., 2025a; Zhang et al., 2024), and facilitate direction interaction with visual inputs through operations like zoom-in (Su et al., 2025; Zheng et al., 2025). Although these extensions can improve interpretability and accuracy, they either inherently add external information, such as auxiliary visual signals and textual rationales, or their applicability is limited to tasks requiring fine-grained visual perception. We propose VAT to facilitate a more direct, visual-centric reasoning process that reduces the reliance on such external resources and auxiliary steps.

**Visual Tool-Using Agents.** In parallel with the development of CoT-based approaches, another promising direction equips models with external visual tools, such as translating visual inputs into verbal format for LLMs (Hu et al., 2022; Yang et al., 2023b), or enhancing MLLMs with external tools for additional visual guidance (Wang et al., 2025a; Zhang et al., 2025a; Zhou et al., 2024). They employ tool-based operations to draw bounding boxes and segment images as a part of intermediate reasoning step for models (Wu et al., 2024; Li et al., 2025a; Hu et al., 2024a), bridging comprehension gaps in tasks such as spatial reasoning (Zeng et al., 2023; Hu et al., 2024b; Gupta & Kembhavi, 2023), and diagram interpretation (Zala et al.; Pan et al., 2024). These approaches help extend verbal explicit thinking into the visual domain, yet often add complexity and are costly. They extend the model capability via external modules and usually add knowledge not present in raw inputs. In contrast, VAT utilizes visual abstract to reorganize existing information in a more reasoning-friendly format for better efficiency, and elicit the inherent model ability to think with these visual abstracts.

**Sketch Representation and Applications.** Building upon early developments in edge-detected sketches (Kittler, 1983; Canny, 1986), recent advances have established deep learning-based sketch representation (Gryaditskaya et al., 2019; Chan et al., 2022; Li et al., 2019) as a notable paradigm for visual understanding. It offers the abstraction and compression of visual context while retaining essential semantic and structural cues. Consequently, research has long focused on uncovering the implicit semantics conveyed through sketches, their synthesis from images or text, and diverse applications such as image retrieval (Song et al., 2017; Pang et al., 2019; Sangkloy et al., 2022; Yu et al., 2016), scene composition (Zou et al., 2018), and integration as auxiliary tools in multimodal reasoning (Vinker et al., 2024; Hu et al., 2024a; Tversky & Tvesky, 1999). Therefore, we employ sketches as our visual abstract to reduce visual noise and emphasize salient structures for MLLMs.

## 3 VISUAL ABSTRACT THINKING (VAT)

Humans demonstrate *visual abstract thinking* by selectively focusing on essential visual elements while disregarding redundant details during visual reasoning. For instance, when counting the number of cars in a photograph, background information is typically ignored. We hypothesize that this cognitive strategy is also beneficial for MLLMs. Usually, visual abstracts retain the primary conceptual, relational and structural visual information of original images, and can be represented as silhouettes, edge features and free-hand sketches. In this paper, To emulate *visual abstract thinking* in MLLMs, we transform input images into sketch, a typical form of visual abstract, extracting core concepts from complex and cluttered scenes. These sketches are then provided as complementary visual perception signals to guide and enhance the reasoning process of MLLMs.

Specifically, we formulate the query for a multimodal task as $Q = [I ; T]$, where $I$ and $T$ refer to image and text input, respectively. $\mathcal{I}$ denotes regular image space, $I \in \mathcal{I}$. Denoting model predicted answer as $A$, the reasoning processes of by CoT and tool-using methods are as follows:

$$\textbf{Chain-of-Thought Reasoning:} \quad Q \rightarrow (T_{\text{chain}}, A), \quad (1)$$

$$\textbf{Tool-using Reasoning:} \quad (Q, V_c^n) \rightarrow A, \quad (2)$$

where $T_{\text{chain}}$ represents a CoT-style intermediate reasoning trace expressed in natural language, which primarily belongs to the verbal reasoning space $\mathcal{L}$. $V_c$ represents a concrete visual trace generated by external tools, such as Grounding DINO (Liu et al., 2024) and SAM (Kirillov et al., 2023)). It belongs to the concrete visual space $\mathcal{V}_c$. Both types of reasoning traces provide detailed content grounded in

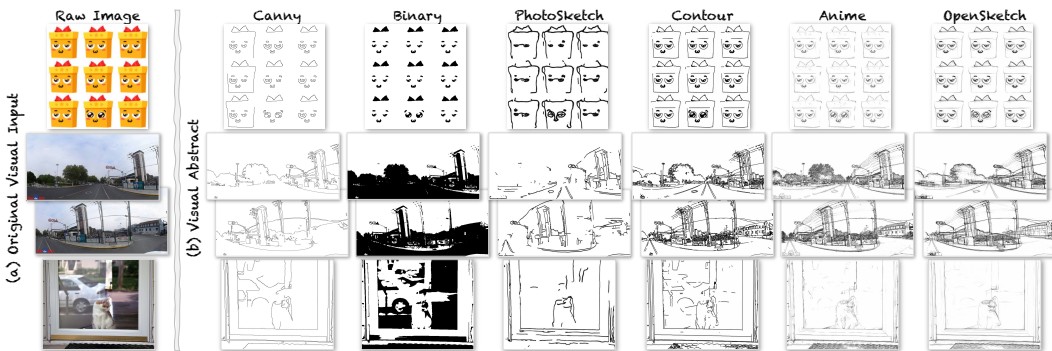

Figure 3: Visualization of different visual abstract types feasible for VAT paradigm, e.g., Canny edge detection (Canny, 1986), image binarization (binary), PhotoSketch (Li et al., 2019), Contour, Anime, and OpenSketch styles (Chan et al., 2022). Images taken from CoSpace and BLINK.

additional linguistic or visual observations. In comparison, our designed VAT is described as follows:

$$\textbf{Visual Abstract Thinking (VAT):}\quad (Q, V_a) \to A. \tag{3}$$

It introduces visual abstraction $V_a \in \mathcal{V}_a$ into the input context, where $\mathcal{V}_a$ denotes the space of geometric, structural and semantic visual abstractions. Unlike explicit visual traces that supply the model with external visual messages, $V_a$ captures internal conceptual and spatial regularities and relations, enabling the model to engage in *visual abstract thinking* from the input image itself. VAT performs reasoning over a joint space $\mathcal{L} \cup \mathcal{V}_a$, integrating verbal reasoning with visual abstraction. The model internally generates reasoning steps in the language space $\mathcal{L}$, while simultaneously attending to aligned semantic and conceptual visual abstractions in $\mathcal{V}_a$, thereby enhancing its perceived visual information and consequently boosting the final reasoning performance.

**Visual Abstract Transformation and Prompt Templates.** To reflect the process of visual abstract thinking for multimodal perception and reasoning tasks, VAT uses image-to-*visual abstract* transform function $T : \mathcal{I} \to \mathcal{V}_a$ to converts input images $I$ into visual abstracts $V_a$, preserving essential visual concepts, geometric information, silhouettes, and relational features. This function $T$ can take diverse forms. In this paper, we employ both code-based conversion approaches, including binary maps and edge-detected outlines, and semantic-based deep learning methods, such as contour-style sketch generation. These abstracted visual representations $V_a$ are incorporated into the multimodal input context to support reasoning with our VAT paradigm described by Equation 3. By transforming complex visual scenes into simplified yet semantically meaningful forms, MLLMs can focus on core concepts and underlying object relationships in visual abstracts ($V_a$), which enhances the performance of visual perception and reasoning tasks via language-abstract joint space $\mathcal{L} \cup \mathcal{V}_a$. Figure 1 demonstrates the prompt of VAT, where we add an visual abstract to the interleaved input without additional CoT-prompting or external visual messages. Please refer to Appendix B for detailed prompt templates of VAT, together with templates used for our employed baselines.

## 4 EXPERIMENTS AND RESULTS

In this section, we validate our VAT across multimodal perception and reasoning tasks, including object-centric perception, and spatial, relation, and commonsense reasoning. We demonstrate the effectiveness of our proposed VAT over both CoT-like reasoning approaches and tool-using approaches. Main experimental results are listed in Table 1, revealing the potential of *visual abstract thinking* in multimodal reasoning tasks. It serves as a complementary approach to *verbal explicit thinking*, offering an alternative perspective that further enriches the overall reasoning landscape.

### 4.1 IMPLEMENTATION

**Representing Images as Visual Abstracts.** We report main experimental results using deep learning-based semantic visual abstractions converted by sketch models trained with both semantic and geometry loss (Chan et al., 2022; Gryaditskaya et al., 2019), which effectively preserve key visual elements such as object structure and overall layout while filtering out irrelevant details, allowing MLLMs to focus on the most relevant visual information for target questions. Figure 3 shows examples across different types of visual abstractions. These approaches focus on various types of visual essence, including Canny, Binary, Photosketch, Contour, and OpenSketch.

Table 1: Main Experimental results comparing VAT with visual and verbal explicit thinking paradigms, Scaffold and CoT, respectively, across different models. We employ OpenSketch converted sketches as the visual abstraction in this table for fair comparison.

| Method | Odd-One-Out | Visual Illusion | Acti View | Text VQA-GT | BLINK | | | | CoSpace | | | | | Avg. Gain |
|---|---|---|---|---|---|---|---|---|---|---|---|---|---|---|
| | | | | | Spatial | Count | Sem.Corr. | Vis.Corr. | Dir-Rec | Dir-Obj | Rot-Ang | Rot-Diff | Count | |
| Qwen2.5-VL-32B | 37.25 | 60.42 | 71.76 | 69.65 | 83.92 | 74.17 | 47.48 | 77.91 | 36.33 | 48.12 | 55.56 | 68.84 | 34.54 | - |
| + Scaffold | 7.84 | 61.81 | 73.73 | 76.35 | 83.22 | 66.67 | 45.32 | 77.91 | 34.80 | 41.20 | 53.33 | 30.50 | 37.00 | -5.87 |
| | -29.41 | +1.39 | +1.97 | +6.70 | -0.70 | -7.50 | -2.16 | +0.00 | -1.53 | -6.92 | -2.23 | -38.34 | +2.46 | |
| + CoT | 41.18 | 58.74 | 72.27 | 60.38 | 81.82 | 62.50 | 46.04 | 67.44 | 38.80 | 42.17 | 57.35 | 73.50 | 31.50 | -2.48 |
| | +3.93 | -1.68 | +0.51 | -9.27 | -2.10 | -11.67 | -1.44 | -10.47 | +2.47 | -5.95 | +1.79 | +4.66 | -3.05 | |
| + VAT (Ours) | 35.29 | 62.50 | 68.75 | 77.00 | 84.67 | 74.17 | 51.08 | 82.56 | 31.82 | 56.80 | 57.33 | 64.80 | 31.41 | +0.94 |
| | -1.96 | +2.08 | -3.01 | +7.35 | +0.75 | +0.00 | +3.60 | +4.65 | -4.51 | +8.68 | +1.77 | -4.04 | -3.13 | |
| Gemini-2.5-Flash | 66.67 | 56.94 | 77.13 | 76.18 | 86.71 | 69.17 | 57.55 | 83.14 | 49.60 | 52.82 | 61.33 | 78.50 | 38.69 | - |
| + Scaffold | 23.53 | 56.94 | 76.74 | 77.93 | 86.71 | 74.17 | 45.32 | 82.56 | 44.98 | 47.60 | 64.33 | 88.38 | 38.00 | -3.63 |
| | -43.14 | +0.00 | -0.39 | +1.75 | +0.00 | +5.00 | -12.23 | -0.58 | -4.62 | -5.22 | +3.00 | +9.88 | -0.69 | |
| + CoT | 62.75 | 58.33 | 71.71 | 75.71 | 85.31 | 72.27 | 58.27 | 88.95 | 61.54 | 62.80 | 61.33 | 84.18 | 42.50 | +2.40 |
| | -3.92 | +1.39 | -5.42 | -0.47 | -1.40 | +3.10 | +0.72 | +5.81 | +11.94 | +9.98 | +0.00 | +5.68 | +3.81 | |
| + VAT (Ours) | 72.55 | 59.03 | 75.58 | 76.40 | 84.62 | 66.10 | 56.12 | 93.02 | 52.21 | 60.24 | 75.59 | 86.87 | 43.50 | +3.65 |
| | +5.88 | +2.09 | -1.55 | +0.22 | -2.09 | -3.07 | -1.43 | +9.88 | +2.61 | +7.42 | +14.26 | +8.37 | +4.81 | |
| GPT-4o | 25.49 | 55.56 | 76.34 | 74.20 | 82.52 | 65.83 | 53.96 | 86.05 | 40.40 | 46.00 | 58.33 | 50.50 | 40.00 | - |
| + Scaffold | 0.00 | 59.72 | 78.52 | 76.05 | 87.41 | 61.67 | 43.18 | 77.33 | 38.40 | 46.00 | 50.00 | 79.00 | 31.00 | -2.07 |
| | -25.49 | +4.16 | +2.18 | +3.42 | +4.89 | -4.16 | -10.78 | -8.72 | -2.00 | +0.00 | -8.33 | +28.50 | -9.00 | |
| + CoT | 29.41 | 54.17 | 78.52 | 77.62 | 84.62 | 72.50 | 66.19 | 88.95 | 43.55 | 43.32 | 57.19 | 77.84 | 47.00 | +5.05 |
| | +3.92 | -1.39 | +2.18 | +3.42 | +2.10 | +6.67 | +12.23 | +2.90 | +3.15 | -2.68 | -1.14 | +27.34 | +7.00 | |
| + VAT (Ours) | 45.10 | 59.03 | 78.63 | 75.85 | 85.31 | 73.33 | 63.31 | 93.6 | 44.00 | 53.60 | 50.33 | 94.50 | 38.50 | +7.69 |
| | +19.61 | +3.47 | +2.29 | +1.65 | +2.79 | +7.50 | +9.35 | +7.55 | +3.60 | +7.60 | -8.00 | +44.00 | -1.50 | |
| GPT-5 | 70.59 | 59.03 | 87.20 | 82.10 | 86.71 | 71.67 | 69.78 | 94.08 | 68.00 | 72.40 | 65.00 | 78.50 | 45.23 | - |
| + Scaffold | 45.10 | 56.94 | 80.08 | 81.43 | 88.11 | 68.33 | 68.61 | 93.02 | 65.20 | 70.80 | 64.00 | 80.00 | 42.13 | -3.58 |
| | -25.49 | -2.09 | -7.12 | -0.67 | +1.40 | -3.34 | -1.17 | -1.06 | -2.80 | -1.60 | -1.00 | +1.50 | -3.10 | |
| + CoT | 61.22 | 60.42 | 85.58 | 85.00 | 87.41 | 78.15 | 71.94 | 93.53 | 66.80 | 79.20 | 63.67 | 83.42 | 46.46 | +0.96 |
| | -9.37 | +1.39 | -1.62 | +2.90 | +0.70 | +6.48 | +2.16 | -0.55 | -1.20 | +6.80 | -1.33 | +4.92 | +1.23 | |
| + VAT (Ours) | 83.67 | 61.81 | 80.47 | 81.50 | 88.03 | 75.83 | 71.94 | 95.86 | 68.00 | 77.60 | 60.33 | 87.00 | 47.00 | +2.21 |
| | +13.08 | +2.78 | -6.73 | -0.60 | +1.32 | +4.16 | +2.16 | +1.78 | +0.00 | +5.20 | -4.67 | +8.50 | +1.77 | |

**Models.** We validate VAT on both proprietary and open-source MLLMs, and examine its generality across model scales, architectures and training recipes. We evaluate the performance of VAT on representative contemporary MLLMs, including proprietary models[1] and open-source models, such as Qwen2.5-VL and MiniCPM-V 2.5. For proprietary models with adjustable reasoning efforts, we investigate the effect of VAT with different thinking budgets.[2]

**Evaluation Tasks** To assess the generality of our proposed VAT, we conduct experiments across diverse tasks of visual perception and multimodal reasoning, spanning both single-image and multi-image settings. In detail, **Object-Centric Perception** focuses on interpreting and analyzing physical attributes of individual objects, where VAT helps isolate core object features. We evaluate this on BLINK (Fu et al., 2024) (count), HallusionBench (Guan et al., 2024) (illusion), TextVQA-GT (Singh et al., 2019) [3], and CoSpace (Zhu et al., 2025) (DIR-Obj). **Object Relation Reasoning** involves understanding semantic mappings and structural relations of objects, that VAT assists in highlighting them. We evaluate this using our created Odd-One-Out benchmark and BLINK (semantic, and visual correspondence). **Spatial Reasoning** includes BLINK (spatial) and CoSpace (DIR-Rec, ROT-Ang, ROT-Dif), which tests the ability to recognize spatial and orientational semantics based on perception of objects and scenes, where VAT provides more explicit sketches of these information. We also investigate ActiView (Wang et al., 2025b) that addresses active perception and covers both the above tasks. Please refer to Appendix C for details of our created Odd-One-Out benchmark and other employed tasks, with corresponding evaluation metrics.

## 4.2 MAIN RESULTS

We report experimental results involving six benchmarks in Table 1. Results demonstrate that VAT consistently outperforms employed baselines, the verbal explicit reasoning paradigm (CoT) and the visual-augmented concrete reasoning method (Scaffold), and is more effective and efficient compared with ReAct-based tool-using methods (as in Table 2).

**VAT is consistently effective across tasks, model sizes, and architectures.** Trends across different model architectures and scales imply that the *visual abstract thinking* paradigm can boost multimodal reasoning performance, including spatial reasoning, object-centric understanding, and relational inference, via abstract perceptual context. In contrast, Scaffold promotes visual-language coordination and performs well on the Visual Illusion task, since its overlaid dot matrix on images helps determine relative object sizes. However, it is less effective on other tasks, where the localization and size

---

[1] Employed API version: GPT-4o: gpt-4o-2024-11-20; GPT-5: gpt-5-2025-08-07; Gemini-2.0-pro: gemini-2.0-pro-exp-02-05; Gemini-2.5-Flash: gemini-2.5-flash

[2] Reasoning parameters: Gemini-2.5-Flash: `thinkingBudget=0`; and GPT-5: `reasoning_effort='medium'`

[3] This is a subset of TextVQA (Singh et al., 2019) with manually annotated groundtruth.

Table 2: Experimental results in ReAct-style frameworks with GPT-4o. This table compares VAT and the other visual tool-using work, Visual SketchPad (Hu et al., 2024a). For fair comparison, we implementing our VAT in ReAct pipeline as SketchPad does, and report the gain of VAT over SketchPad. OpenSketch is used for visual abstract transformation. ↓ denotes degraded results compared to GPT-4o. Appendix D provides more discussions of implementing VAT with ReAct.

| Method | BLINK | | | | CoSpace | | | | |
|---|---|---|---|---|---|---|---|---|---|
| | Spatial | Count | Sem.Corr. | Vis.Corr. | Dir-Rec | Dir-Obj | Rot-Ang | Rot-Diff | Count |
| GPT-4o | 82.52 | 65.83 | 53.96 | 86.05 | 40.40 | 46.00 | 58.33 | 50.50 | 40.00 |
| Visual SketchPad | 82.42↓ | 68.32 | 57.97 | 80.23↓ | 40.80 | 40.96↓ | 50.67↓ | 84.83 | 25.45↓ |
| **VAT**-ReAct (Ours) | 85.11 +2.69 | 69.17 +0.85 | 62.32 +4.35 | 91.28 +11.05 | 46.59 +5.79 | 49.60 +8.64 | 51.00↓ +0.33 | 89.50 +4.67 | 34.85↓ +9.40 |

perception at which Scaffold excels are less critical for those tasks. While CoT, an explicit thinking method, enhances multimodal reasoning ability for most models, its performance gain diminishes as newer models are post-trained with stronger inherent reasoning abilities (e.g., dropping from a +5.05% gain on GPT-4o to +0.96% on GPT-5). The performance gap between CoT and VAT suggests that enhancing perception by filtering redundant information remains a promising direction.

**Comparison with typical tool-using method.** We compares VAT with representative multimodal tool-using method Visual Sketchpad (Hu et al., 2024a). For fair comparison, we implement VAT-ReAct following ReAct-style pipeline (Yao et al., 2023b). Table 2 lists experimental results on GPT-4o, showing that VAT obtains an average improvement of 5.31% over this widely used framework, with notable advantages in semantic correspondence (11.05%) of BLINK, and Dir-Obj (8.64%) and Count (9.40%) tasks of CoSpace, further demonstrating the effectiveness of visual abstracts in multimodal perception and reasoning tasks regarding different implementations. We refer readers to Appendix D for more results of VAT-ReAct, including Gemini-2.0-Pro.

These results collectively suggest that the *visual abstract thinking* paradigm offers substantial advantages via abstract perception format. While CoT and Scaffold, representing *verbal explicit reasoning* and *visual-augmented concrete reasoning* respectively, contribute valuable perspectives, VAT provides a complementary approach that emphasizes conceptual abstraction and minimalism, fostering a more efficient and focused visual perception and reasoning process.

## 4.3 ABLATION STUDY

Experimental results demonstrate the effectiveness of our proposed VAT, and exhibit superiority against other verbal explicit thinking and visual guidance paradigms, including CoT (Wei et al., 2022) and Scaffold (Lei et al., 2025), in multimodal perception and reasoning tasks. To further investigate the underlying mechanism of VAT, we design ablation studies in this section regarding prompt templates and the provided visual information. Specifically, we vary the prompt template, progressively supply models with visual abstractions to observe the influence of significant visual information, and calculate the corresponding trends of correctly predicted tokens.

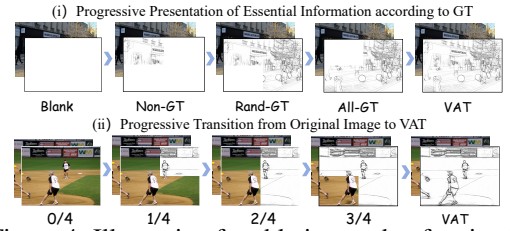

Figure 4: Illustration for ablation study of maintaining or filtering visual information via sketches.

Table 3: Results of ablation study depicted in Figure 4. From "Non-GT" (presenting no GT regions) to "All-GT" (preserving all GT regions via sketch), the volume of visual information conveyed by sketch increases.

| Benchmarks | Baseline (Img-only) | Present Regions | | | VAT |
|---|---|---|---|---|---|
| | | Non-GT | Rand-GT | All-GT | |
| Actiview | 76.34 | 75.19 | 77.09 | 77.86 | **78.63** |
| TextVQA-GT | 74.20 | 75.45 | 75.75 | **76.23** | 75.85 |

**How visual abstract assists in reasoning?** Towards understanding the underlying mechanism of VAT, we conduct ablation studies on two benchmarks, ActiView (Wang et al., 2025b) and TextVQA-GT (Zhang et al., 2025b), as they provide manually annotated ground-truth bounding boxes (BBox), indicating visual information required by human to answer corresponding questions. Following Wang et al. (2025b), we divide images into equal blocks, assigning a label to each of them to indicate if it contains the ground-truth (GT) BBox. Then, we employ two settings in Figure 4 for investigation: (i) progressively present visual abstract to models via VAT according to the GT information, and (ii)

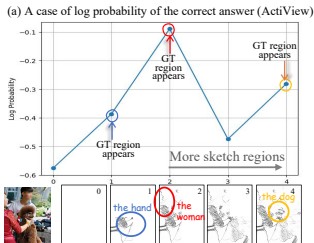 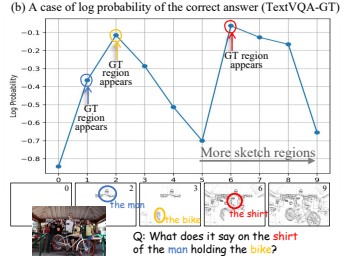 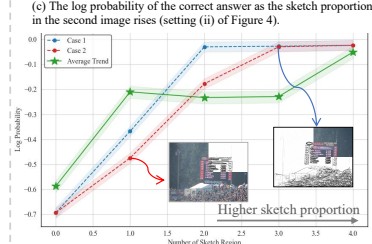

Figure 5: The effect of visual abstract on the probability of predicted answer tokens by GPT-4o. x-axis: the number of presented sketch regions. In (a) and (b) sketch regions are randomly added to a blank base. (c) contains the overall trend (green) and two cases presented by setting (ii) in Figure 4.

gradually transform image regions into visual abstract. Through these configurations, we examine if VAT functions similarly to selectively screening redundancy while retaining essential cues.

The results of first ablation study, setting (i), are shown in Table 3. We observe an ascending trend in final accuracy, from "Non-GT" to "All-GT", where visual abstract of the GT regions progressively appear. This demonstrates that sketch can preserve truly required visual information, which consequently improves the final perception and reasoning performance. We provide additional cases to randomly present GT regions with sketch in Figure 5 (a) and (b), showing that the probability of the correct answer increases with the appearance of relevant ground-truth regions, such as the dog in (a) and the shirt in (b), while it decreases with the presence of redundant information.

Whilst setting (i) discusses the preserving of required information via visual abstract from the input side, setting (ii) addresses the effect on the output side. In Figure 5 (c), the overall trend shows that gradually transitioning the image into visual abstract increases the log probability of generating the correct answer, compared to simply providing two identical images. Furthermore, we showcase different order of the transition regarding the GT region. We observe that when sketching out the redundant regions prior to the GT regions (blue line in Figure 5 (c)), the trend of probability is steeper than the GT-first order (red line in Figure 5 (c)). This observation supports our hypothesis that as an abstract form, sketch effectively filters out redundancy and enhances performance.

**Ablation Study on VAT Prompting Formats.** Since models are often sensitive to prompting formats, we conduct ablation study to compare different visual input configurations. VAT employs a dual-image prompting template, with a raw image $I$ and a visual abstract $V_a$. We consequently discuss the influence of single-image and dual-image formats via three configurations for each, and list the corresponding average results. We refer readers to Appendix G

Table 4: Ablation study of prompting formats on GPT-4o with OpenSketch style abstract. We calculate the average scores of our employed sub-tasks of Blink, CoSpace, Illusion and Odd-One-Out in Table 1.

| Single-image Format | | | Dual-image Format | | |
|---|---|---|---|---|---|
| Blank | Img | Vis.Abs | VAT w/ Blank | VAT w/ Img | **VAT(Ours)** |
| 27.47 | 52.51 | **54.51** | 55.73 | 60.41 | **63.09** |

for discussion over tasks. For single-image format, we test the performance of feeding a white image as placeholder (Blank), $I$ (Img), and $V_a$ (Vis.Abs) in the input, respectively. For dual-image format, we supply the raw image $I$ with additional blank placeholder (VAT w/ Blank), $I$ (VAT w/ Img, repeating $I$), and $V_a$ (VAT, $I$ together with $V_a$), respectively following the single-image ablation.

Results are shown in Table 4. The blank placeholder results in only 27.47% accuracy, implying that these tasks require participation of visual information, and cannot be solved merely through text. The highest average score of single-image format is presented by "Vis.Abs", supporting the effectiveness of abstract visual perception. Although dual-image results illustrate that additional image can improve the performance thought over single-image format, visual abstract $V_a$ further assists in guiding models to necessary information and leading to the correct answer (63.09% *vs* 60.41% and 55.73%). This study shows that the improvement in VAT does not come simply from more visual inputs, indicating the potential of visual abstract in tasks involving massive visual context.

## 5 ANALYSIS

In this section, we conduct additional experiments to figure out the impact of VAT across different tasks and models, and the flexibility regarding abstract styles. We also discuss its effect on responding efficiency. Case studies of VAT can be found in Appendix K.

## 5.1 IMPACT ACROSS TASKS AND MODELS

While VAT consistently improves task performance, the gain achieved by VAT varies across task types, as shown in Figure 6. VAT benefits visual reasoning through visual abstracts that stress implicit structural and semantic cues, stimulating abstract thinking from the visual side. It is particularly helpful for coarse-grained tasks, such as object relation reasoning (Odd-one-out, BLINK-Sem-corr, etc) and spatial reasoning (BLINK-Spatial, CoSpace-Rot tasks, etc), where VAT can effectively assists in filtering out irrelevant background clutters. However, in comparison to these tasks, VAT sometimes provides obscure details via visual abstract, limiting the performance gain for some object-centric tasks requiring fine-grained perception, such as counting tasks and HallusionBench-Illusion.

*TAKEAWAY: VAT benefits from visual abstracts that highlight essential visual elements while blurring the clutters, especially for object relation and spatial reasoning tasks.*

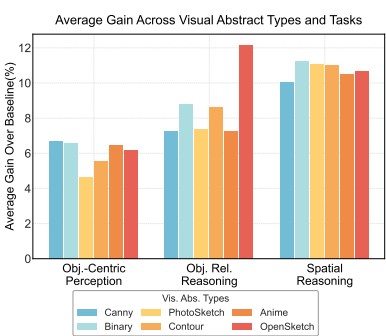

While all models benefit from VAT, the extent of improvement varies considerably across model size and training recipe. As reported in Table 1, Appendix E, and Appendix I.2, larger scale proprietary models, such as GPT-4o and Gemini-2.0-Pro, exhibit the most substantial gain, achieving relative improvements of 52.27% and 21.51% over corresponding CoT baselines, respectively. While the improvements are less notable for smaller opensource models, such as Qwen2.5-VL-32B and -7B models, and for reasoning-augmented models, such as GPT-5. This disparity may stem from the model scale and verbal-thinking enhanced training recipes, which makes models to better utilize verbal thinking instead of visual thinking.

Figure 6: Visualization of performance gain of VAT over baseline (across visual abstract styles).

*TAKEAWAY: VAT consistently boost performance for models of different scales and thinking modes, but it advances more for non-reasoning models, comparing to models optimized for verbal reasoning.*

## 5.2 INFLUENCE OF ABSTRACT STYLES COMBINATION AND SELECTION

Intuitively, features and information vary across different visual abstract styles. To further investigate the potential benefits of incorporating multiple abstract types, we conducted an experiment to simultaneously provide multiple distinct styles. We report the average performance in Table 5, and provide detailed results in Appendix J. First of all, Our VAT achieves the highest average accuracy and gain with single style (+8.72), OpenSketch, compared with combinations of two (+6.15) and three (+4.79 and +3.89) styles. This demonstrate that integrating more information do not necessarily produce better results, which further support our motivation to enable concentrated reasoning via visual abstract. Notably, among the experimental results of combining three styles, Combo A outperforms Combo B, in that Combo A covers largely distinct visual abstract styles while Combo B uses similar styles which potentially increases the redundancy issue.

Table 5: Average results (GPT-4o) of style and selection. 2-Style Combo: Binary + OpenSketch; 3-Style Combo A: Binary + PhotoSketch + Opensketch; 3-Style Combo B: Aminie + Contour + Opensketch. VAT (Rule-based): rule-based selection of only one style. Detailed tasks and results are in Table 13.

| Combinations | Avg. ACC | Avg. Gain |
|---|---|---|
| Img-only | 54.97 | - |
| VAT (OpenSketch) | 63.69 | +8.72 |
| VAT (Rule-based) | **64.72** | **+9.75** |
| 2-Style Combo | 61.12 | +6.15 |
| 3-Style Combo A | 59.76 | +4.79 |
| 3-Style Combo B | 58.86 | +3.89 |

Based on results in Section 5.1, we validate a straightforward approach to further enhance the performance via abstract style selection beyond static styles. As shown in Figure 3, different abstract styles contribute differently regarding tasks. We employ a rule-based selection prompt that direct models to different style according to the target tasks. Results are reported as VAT (Rule-based) in Table 5 and 13. Simple rule-based selection achieves average higher accuracy of 64.72%, compared with VAT with only OpenSketch (63.69%), implying that the performance can be further improved.

*TAKEAWAY: Single abstract style can effectively and efficiently improve the performance, while multiple styles potentially increase redundancy.*

## 5.3 Analysis of Efficiency

A key advantage of VAT is its efficiency, because it enhances reasoning by adding information to the input rather than relying on the generation of verbose, costly rationales in the output. This is particularly significant given that output tokens are often far more expensive than input tokens, with the price ratio for models like Gemini-2.5-Flash being as high as 8:1.

We compare the efficiency of VAT to verbal and visual explicit thinking approaches (CoT and Sketchpad) on GPT-4o, focusing on token cost and runtime[4] across object-centric perception and relational reasoning tasks, including odd-one-out, illusion, and count and semantic correspondence of BLINK. As visualized in Figure 2 and Figure 7, VAT achieves higher accuracy than CoT-style verbal explicit reasoning, with only $34.72\%$ increase in token consumption for these tasks. Compared to tool-using methods, Visual Sketchpad, who costs notably $13.46\times$ CoT, $20.96\times$ Baseline, and $9.99\times$ VAT, VAT significantly reduces token cost[5] while consistently delivering superior performance. Moreover, VAT is also time-efficient compared to explicit thinking methods CoT and Visual SketchPad, requiring only $67.62\%$ and $27.53\%$ of the runtime compared to them, respectively.

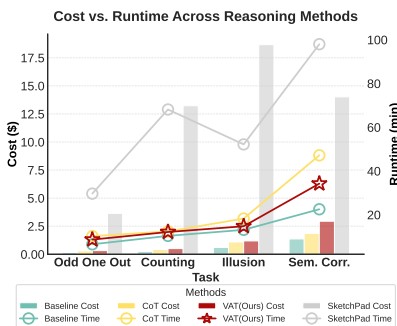

Figure 7: Runtime and token cost comparison across methods.

Beyond discussion over typical task above, we further evaluate the efficiency of VAT on Gemini-2.5-Flash over all investigated tasks to derive a generalizable conclusion. The results, shown in Table 6, reflect its performance across all benchmarks from Table 1. This study does not include Visual Sketchpad due to budget constraints. Given that API latency can cause variability in runtime measurements, and that output

Table 6: Total cost and token consumption for tasks listed in Table 1. Costs are based on official pricing of Gemini API, and Sum Tokens are the sum of inputs and outputs.

| Metric | Standard | Scaffold | CoT | VAT (Ours) |
|---|---|---|---|---|
| *Gemini-2.5-Flash* | | | | |
| Cost ($) | 2.67 | 6.15 ×2.3 | 11.420 ×4.3 | **5.70** ×2.1 |
| Output Tokens | 199.95 | 819.45 ×4.1 | 1,481.95 ×7.4 | **538.28** ×2.7 |
| Sum Tokens | 1,142.80 | **1,825.12** ×1.6 | 2,431.88 ×2.1 | 2,141.30 ×1.9 |

tokens are usually more expensive than input tokens, we used the output token count as an alternative reflection of efficiency regarding time and cost. The results show that VAT consumes the lowest total cost and requires the fewest output tokens comparing to Scaffold and CoT, conclusively demonstrating its superior efficiency.

*TAKEAWAY: VAT enhances accuracy while reducing token consumption and runtime by utilizing low-cost input tokens for reasoning, rather than relying on reasoning with expensive generate tokens.*

## 5.4 The Potential of Test-time Scaling

As shown in Table 1, we observe that Qwen-2.5-VL-32B ($35.29\%$) initially underperforms standard prompting ($37.25\%$) on the Odd-One-Out task with a temperature of 0. However, applying test-time scaling at a temperature 0.9, the model achieves a Pass@5 of $49.02\%$, significantly surpassing the standard prompting baseline ($45.10\%$). This result suggests that test-time scaling can furthre enhance the performance of VAT, offering a prospective direction of VAT for a broader investigation across tasks and for future research as well.

## 6 Conclusion

In this paper, we introduce ***visual abstract thinking***, and propose a new multimodal reasoning paradigm, **V**isual **A**bstract **T**hinking (**VAT**). Experimental results and cases demonstrate that VAT utilizes visual abstracts, simplified and concise representations of images, to convey essential visual elements out of massy and redundant context. It outperforms investigated verbal and visual prompting methods, and achieves remarkably higher time- and cost-efficiency compared with tool-using methods. VAT also presents good flexibility and generality. We hope this study could urge exploration of more diverse, effective, and efficient reasoning paradigms from the perspective of human cognition.

---

[4]Runtime measurements are subject to variability due to API latency and may not reflect absolute accuracy.

[5]API pricing is based on https://openai.com/api/pricing/

**Limitation and Future Work**  This work discusses VAT for academic benchmarks, but future research should also explore its applicability to real-world tasks, such as GUI tasks and in embodied scenarios, which present challenges beyond merely perception and reasoning. Moreover, the internal information flow within models caused by VAT is also a promising topic to investigate.

## REPRODUCIBILITY STATEMENT

The prompts, detailed API versions, and parameter settings necessary for reproduction are provided in Section 4, and through Appendix B to E, including details for both the main experiments, ablation studies and additional analysis. The code used to generate our results is submitted within the supplementary materials.

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

## A    LLM USAGE STATEMENT

Throughout the completion of this work, LLMs are solely used for the purpose of spelling checking and grammatical error detection for the manuscript. They do not participate in deriving idea of this paper, nor designing the methods or validating the results.

## B    PROMPT TEMPLATES

### B.1    PROMPT TEMPLATES FOR VAT

Detailed prompt templates of all tasks for employed proprietary models are as follows:

```
<question> // Query-image input and original task instructions
<sketch of images> // Visual Abstract of original image(s)
/* Instruction of VAT */
Each image is provided together with an visual abstract converted from
the original image.  It helps you determine essential components in the
image, including but not limited to spatial, structural, relational,
and conceptual features, which can assist in reasoning.  You should use
both the original image and the sketch to inform your reasoning process.
Sketches are really useful, you must fully utilize them to achieve the
best performance.  You should follow the format:  ANSWER: (your answer).
E.g.:  ANSWER: (A). Your answer:
```

### B.2    PROMPT TEMPLATES FOR BASELINES METHODS

In this section, we provide standard prompt templates for models without applying anything reasoning-enhancing instructions, as well as prompt templates for Scaffold and extended VAT+CoT.

**Standard Prompt Template**    Standard prompt template for investigated models is shown as follows:

```
<question> // Query-image input and original task instructions
/* Instruction of Standard Prompt */
Please reply in the following format:
ANSWER: (your answer).  Example:  ANSWER: (A) Your response:
```

**Prompt Template for Scaffold**    Prompt Template of Scaffold is shown as follows:

```
<question> // Query input without image and original task instructions
<overlaid image> // Image(s) with Scaffolding Coordinates
/* Instruction of Scaffold */
Each image will be provided together with a corresponding sketch, which
is directly converted from the original image.  The sketch helps you
determine essential components in the image, including but not limited
to spatial, structural, relational, and conceptual features, which can
assist in your reasoning process.  You should use both the original
image and the sketch to inform your reasoning process.  Sketches
are really useful, you must fully utilize them to achieve the best
performance.
You should reply in the following format:  ANSWER: (your answer).  For
example:  ANSWER: (A). Your answer:
```

**Prompt Template for VAT + CoT**    Prompt Template of VAT + CoT, enabling additional chain-of-thought reasoning for VAT, is shown as follows:

```
<question> // Query-image input and original task instructions
<sketch of images> // Visual Abstract of original image(s)
/* Instruction of VAT + CoT */
Each image is provided together with an visual abstract converted from
the original image.  It helps you determine essential components in the
image, including but not limited to spatial, structural, relational,
```

```
and conceptual features, which can assist in reasoning.  You should use
both the original image and the sketch to inform your reasoning process.
Sketches are really useful, you must fully utilize them to achieve the
best performance.  You should reply in the following format:
ANSWER: (your answer).  For example:  ANSWER: (A). Please think step by
step to obtain the answer:
```

## C  BENCHMARKS AND EVALUATION METRICS

### C.1  CONSTRUCTION OF ODD-ONE-OUT BENCHMARK

*Odd-One-Out* is our newly constructed benchmark designed to evaluate the capability of multimodal large language models in identifying the distinct object within an $m \times n$ grid, as shown in Figure 1. The data is sourced from an online game titled *Odd One Out*[6]. The dataset comprises a total of 51 tasks, where both $m$ and $n$ range from 3 to 5, providing a balanced and effective assessment of visual discrimination abilities.

### C.2  EVALUATION METRICS

We extract the answer from model output by pattern matching. For all tasks except MME, we compute accuracy (ACC) by Equation 4. We adopt sub-string matching as $f_{correct}$ to determine correctness.

$$ACC = \frac{1}{n} \sum_{i=1}^{n} f_{correct}(pred_i, gt_i) \tag{4}$$

The predicted answer ($pred$) is extracted from the raw output via pattern matching, whereas the ground truth is denoted as $gt$.

## D  VAT IN REACT PIPELINE

Table 7: Experimental results in ReAct-style frameworks with GPT-4o and Gemini-2.0. This table compares VAT and the other visual tool-using work, Visual SketchPad(SketchPad). For fair comparison, we implementing our VAT in ReAct pipeline as SketchPad does. OpenSketch is employed for visual abstract transformation.

| Method | BLINK | | | | CoSpace | | | | |
|---|---|---|---|---|---|---|---|---|---|
| | Spatial | Count | Sem.Corr. | Vis.Corr. | Dir-Rec | Dir-Obj | Rot-Ang | Rot-Diff | Count |
| GPT-4o | 82.52 | 65.83 | 53.96 | 86.05 | 40.40 | 46.00 | 58.33 | 50.50 | 40.00 |
| + SketchPad | 82.42 | 68.32 | 57.97 | 80.23 | 40.80 | 40.96 | 50.67 | 84.83 | 25.45 |
| + **VAT**-ReAct (Ours) | **85.11** | **69.17** | **62.32** | **91.28** | **46.59** | **49.60** | **51.00** | **89.50** | **34.85** |
| Gemini-2.0 | 85.11 | 68.07 | 65.44 | 76.47 | 61.13 | 68.80 | 85.08 | 81.54 | 46.00 |
| + SketchPad | 84.62 | 70.94 | 56.15 | 87.21 | 61.72 | 61.63 | 77.09 | 82.65 | 20.34 |
| + **VAT**-ReAct (Ours) | **88.81** | **71.67** | **64.75** | **93.02** | **62.10** | **63.60** | **90.73** | **84.62** | **40.91** |

To ensure a fair comparison, we implemented VAT within a ReAct-style framework, mirroring the tool-augmented reasoning setup used by Visual SketchPad. As shown in Table 7, across both GPT-4o and Gemini-2.0, our **VAT-ReAct** consistently outperforms SketchPad across a wide range of tasks in the *BLINK* and *CoSpace* benchmarks.

In the *BLINK* benchmark, which includes tasks involving spatial reasoning, counting, and object-relational understanding, VAT-ReAct demonstrates superior accuracy across all subtasks. This suggests that even in a reactive tool-calling environment, the visual abstracts generated by VAT offer more semantically aligned representations than the explicit visual cues appended by visual tools. In particular, for the *BLINK - Vis.Corr.* task, VAT-ReAct achieves a substantial performance gain over SketchPad (+11.05 for GPT-4o, +5.81 for Gemini), highlighting the advantage of geometric-semantic abstraction over visual explicit information generated by visual tools.

---

[6]https://gamesnacks.com/games/oddoneout

In the *CoSpace* benchmark, **VAT-ReAct** consistently outperforms SketchPad across all task types, demonstrating its strong generalization in visual reasoning. For spatial reasoning tasks, VAT-ReAct achieves significant improvements, including +5.79 in *Dir-Rec* (46.59 vs. 40.80) and +4.67 in *Rot-Diff* (89.50 vs. 84.83). In object-centric perception (*Dir-Obj*), VAT-ReAct leads by +8.64 (49.60 vs. 40.96), highlighting its effectiveness in capturing object position semantics. Even in the more challenging object-relational reasoning task *Rot-Ang*, it maintains a slight edge (51.00 vs. 50.67), reflecting its robustness. These results confirm that, unlike SketchPad which relies on visual concretes from external tools, semantically-geometric visual abstracts in VAT enable more holistic and context-aware reasoning through the *visual abstract thinking* paradigm.

# E  PERFORMANCE ON OPEN-SOURCE MODELS

To examine the effectiveness and generality of our proposed VAT, apart from Qwen-2.5-VL-32B reported in Table 1, we conduct additional experiments on smaller open-source models of ~7B and ~8B, including Qwen2.5-VL-7B and MiniCPM-V 2.6, respectively. Results are listed in Table 8.

Table 8: Experimental results of VAT applied to smaller open-source models (MiniCPM-V 2.6 and Qwen2.5-VL-7B), using OpenSketch-style visual abstracts.

| Method | Odd-One -Out | Visual Illusion | BLINK | | | | CoSpace | | | | |
| | | | Spatial | Count | Sem.Corr. | Vis.Corr. | Dir-Rec | Dir-Obj | Rot-Ang | Rot-Diff | Count |
| --- | --- | --- | --- | --- | --- | --- | --- | --- | --- | --- | --- |
| MiniCPM-V 2.6 | 7.84 | 58.04 | 80.42 | 63.33 | 33.09 | 45.35 | 32.80 | 40.40 | 50.00 | 56.00 | 38.50 |
| + CoT | **17.65** | **58.33** | 69.93 | 52.50 | 37.41 | 40.70 | 27.31 | 33.06 | **51.18** | 27.50 | 30.00 |
| + VAT | 15.69 | 58.04 | **76.92** | **57.14** | **38.41** | **44.19** | 27.42 | **34.94** | 49.00 | 23.81 | **30.37** |
| Qwen2.5-VL-7B | 9.80 | 60.42 | 82.52 | 65.00 | 39.57 | 55.23 | 25.60 | 36.44 | 35.33 | 62.00 | 37.50 |
| + CoT | **15.69** | 50.00 | 65.73 | 59.17 | 35.25 | 59.30 | 33.33 | 36.11 | **51.51** | **57.00** | 34.50 |
| + VAT | 5.88 | **61.81** | **80.42** | **69.17** | **43.88** | **72.67** | **36.95** | **46.79** | 49.67 | 34.50 | **35.68** |

**Analysis of Open-Source Model Performance**  Compared with CoT, VAT consistently performs better for both models in BLINK tasks that intensively require visual perception. This find also holds for visual, such as those require to recognize directional features, and to count specific objects. However, we notice a significant performance gap bewteen CoT and VAT in rotation-based reasoning tasks of CoSpace.

This performance difference may be explained by the varying cognitive demands of different task types. Visual perception tasks benefit from the semantic simplification provided by VAT, which enables even smaller models to better understand geometric structures and spatial layouts. For example, in tasks such as *Spatial*, *Count*, and *Vis. Corr.*, VAT provides clear improvements over CoT by enhancing the ability of models to extract and align key visual signals.

In contrast, task such as *Odd-One-Out* place emphasis on verbal logical inference, where CoT tends to perform better. One possible explanation is that VAT introduces additional visual abstract images, significantly increasing the input context length. For smaller models like MiniCPM and Qwen2.5-VL, this expanded input may strain their limited attention and processing capacity. Moreover, these models are often pretrained or finetuned with a strong emphasis on CoT-style verbal reasoning, which may make it more difficult for them to adapt to the VAT paradigm that relies on processing and integrating abstract visual representations. As a result, VAT can be less effective in tasks that depend heavily on structured verbal reasoning in these smaller models.

Larger models such as GPT-4o and Gemini demonstrate emergent capabilities in *visual abstract thinking*. These models are able to process and integrate abstract visual representations more effectively, which supports coherent and structured multimodal reasoning.

# F  DISCUSSION ON PERFORMANCE OF REASONING MODELS

We evaluate the effectiveness of VAT on reasoning models under different reasoning-effort settings, as shown in Table 9 and Table 1. For GPT-5, we adopt the default setting `reasoning_effort='medium'`, while for Gemini-2.5-Flash we disable reasoning with `thinkingBudget=0`. By default, Gemini-2.5-Pro automatically allocates a reasoning budget; we

Table 9: Experimental results of VAT on models with different reasoning effort or budget . We employ OpenSketch converted sketches as the visual abstraction in this table for fair comparison.

| Method | Odd-One-Out | Visual Illusion | BLINK | | | | Avg. Gain |
| | | | Spatial | Count | Sem.Corr. | Vis.Corr. | |
|---|---|---|---|---|---|---|---|
| Gemini-2.0-Pro | 68.63 | 56.92 | 85.11 | 68.07 | **65.44** | 76.47 | - |
| + Scaffold | 29.41 -39.22 | 56.52 -0.40 | 88.11 +3.00 | 74.58 +6.51 | 54.01 -11.43 | 84.76 +8.29 | -5.54 |
| + CoT | **78.43** +9.80 | 61.27 +4.35 | 85.71 +0.60 | **77.31** +9.24 | 64.75 -0.69 | 90.70 +14.23 | +6.26 |
| + **VAT** (Ours) | 74.51 +5.88 | **68.50** +11.58 | **89.13** +4.02 | 70.91 +2.84 | 64.96 -0.48 | **94.61** +18.14 | +7.00 |
| Gemini-2.5-Pro (budget=default) | 72.55 | 60.14 | 86.01 | 77.31 | **68.35** | **95.32** | - |
| + Scaffold | 44.44 -28.11 | 60.14 +0.00 | 87.41 +1.40 | 77.31 +0.00 | 61.15 -7.20 | 88.89 -6.43 | -6.72 |
| + CoT | 78.00 +5.45 | **62.96** +2.82 | 90.21 +4.20 | **80.67** +3.36 | 66.42 -1.93 | 91.12 -4.20 | +1.62 |
| + **VAT** (Ours) | **78.43** +5.88 | 59.44 -0.70 | **90.91** +4.90 | 78.81 +1.50 | 66.18 -2.17 | 94.77 -0.55 | +1.48 |
| Gemini-2.5-Pro (budget=128) | 76.47 | 56.94 | 88.11 | 75.83 | **71.94** | 92.44 | - |
| + Scaffold | 43.14 -39.22 | **66.67** +9.73 | 89.51 +3.00 | 79.17 +3.34 | 63.31 -11.43 | 90.12 +8.29 | -4.97 |
| + CoT | 74.51 -1.96 | 62.50 +5.56 | 90.21 +2.10 | **80.83** +5.00 | **71.94** +0.00 | 90.70 -1.74 | +1.49 |
| + **VAT** (Ours) | **80.39** +3.92 | 61.11 +4.17 | **91.61** +3.50 | 77.50 +1.67 | 70.50 -1.44 | **96.51** +4.07 | +2.65 |

therefore test both its default configuration and a minimum setting of `thinkingBudget=128`. Gemini-2.0-Pro does not expose adjustable reasoning parameters.

Results show that all models benefit from VAT. However, in Table 1, the absolute gain is smaller on GPT-5 (+2.21) compared to Gemini-2.5-Flash (+3.65), since GPT-5 is trained with stronger default reasoning capabilities, whereas reasoning is disabled in Gemini-2.5-Flash. We attribute this to the compatibility of VAT with chain-of-thought (CoT) reasoning: VAT enhances CoT by boosting perception from *visual abstracts*. When reasoning strength is already high (e.g., GPT-5 with medium effort), the marginal benefit of VAT diminishes, leading to smaller improvements. Nevertheless, the performance gap between CoT and VAT remains stable across settings, with relative gains of +1.25 on GPT-5 and +1.45 on Gemini-2.5-Flash. This consistency supports our claim that VAT is broadly compatible with state-of-the-art reasoning models.

For Gemini-2.5-Pro, we further observe that allocating a relatively large reasoning budget leads to a degradation in VAT performance, suggesting that, effective though, excessive reasoning effort may interfere with the benefits of perceptual abstraction.

## G   DISCUSSION ON VISUAL INPUTS OF VAT PROMPTING TEMPLATE

In addition to Table 4 Section 4.3, we provide some typical cases of different tasks to further investigate the influence of visual prompting formats of our proposed VAT. Since models are often sensitive to prompting formats, we conduct ablation study to compare several prompting configurations. VAT employs a dual-image prompting template, with a raw image and a visual abstract. Therefore We discuss the influence of single-image and dual-image formats, and list some results in Table 4 due to page

Table 10: Ablation study of VAT prompting formats on GPT-4o. OpenSketch style is employed for this study. HB*: HallusionBench.

| Tasks | Single-image Format | | Dual-image Format | |
| | Img-only | Vis.Abs-only | Img-repeat | VAT(Ours) |
|---|---|---|---|---|
| Odd-one-out | 25.49 | 31.37 | 39.22 | **45.10** |
| HB* - Illusion | 55.56 | 56.94 | 58.33 | **59.03** |
| Blink - Sem.Corr | 53.96 | 44.60 | 58.99 | **63.31** |
| Blink - Spatial | 82.52 | 81.82 | **87.41** | **87.41** |
| CoSpace - Rot-dif | 50.50 | **95.00** | 86.21 | 94.50 |
| CoSpace - Dir-Rec | 40.40 | 45.35 | 43.60 | **45.60** |

limitation. *Img-only* and *Vis.Abs-only*, belonging to the single-image format, investigate prompting with only the raw query image and only the visual abstract image, respectively. *Img-repeat* retains the two-image input, but replaces the visual abstract by the raw image, i.e., showing the original image in the input twice. *VAT (Ours)* represents the full VAT setup, incorporating both the original image and corresponding visual abstraction. This comparison allows us to isolate the impact of prompt composition of VAT on visual reasoning accuracy. Overall, dual-image format outperforms single-image format for most tasks, and VAT further outperforms repeating the raw images, showing

that the improvement of VAT does not merely come from more visual inputs. Moreover, we observe exceptions CoSpace tasks requiring at least four query images, where *Vis.Abs-only* shows better results against *Img-repeat*. These demonstrates the potential of visual abstract in tasks involving massive visual context.

# H    DISCUSSION ON VISUAL ABSTRACTS REGARDING ITS ABILITY TO FILTERING PERCEPTUAL INFORMATION

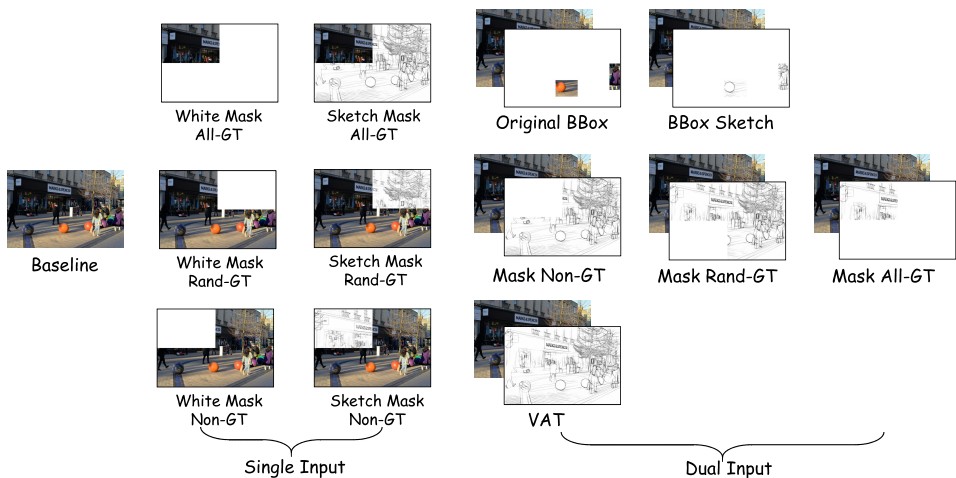

Figure 8: Illustration of the configuration for ablation studies. The single input section on the left shows the model receiving one image, with some regions either masked by a white or sketch mask, which may or may not contain the ground truth, as described in Table 11. On the right, the dual input section presents the model with both the original image and its variants, as detailed in Table 12.

Table 11: Results of single-image input, comparing different masking strategies: (i) presenting all GT regions, (ii) randomly presenting one GT region, and (iii) presenting Non-GT regions. The "White" method replaces the corresponding region with a blank patch, while the "Sketch" substitutes it with the sketch. Simply masking redundant information (e.g., "White") does not necessarily improve performance. And the Sketch method outperforms the White one, highlighting the significance of sketch retaining essential semantic and structural details.

| Method | Baseline | Present Non-GT | Present Rand-GT | Present All-GT |
|--------|----------|----------------|-----------------|----------------|
| White  | 76.34    | 69.08          | 63.08           | 75.95          |
| Sketch | 76.34    | 76.72          | 71.37           | 77.86          |

Table 12: Results of dual-image input (Original + auxiliary image) with similar presenting strategies applied. Here, "BBox" indicates that only the regions containing ground-truth objects are presented, and Sketch-BBox implies that it is derived from the sketch version of the image. Most configurations surpass the baseline, with performance improving as more relevant regions are provided.

| Dataset | Baseline | W/ Region Presents | | | W/ Groundtruth Regions | | VAT |
|---------|----------|--------|---------|--------|------------|---------------|-----|
| | | All-GT | Rand-GT | Non-GT | Sketch-BBox | Original-BBox | |
| Actiview | 76.34 | 77.86 | 77.09 | 75.19 | 77.73 | 78.52 | **78.63** |
| TextVQA-GT | 74.20 | 76.23 | 75.75 | 75.45 | 77.12 | **79.10** | 75.85 |

We examine different configurations involving either a single input or dual inputs with the original image and its visual abstract, the sketch. In the single-input setting, the region of interest is presented in original or sketch form. In the dual-input setting, we supplement the original image with additional information. This includes supplying only the ground truth bounding box, providing the region that does or does not contain the ground truth in original or sketch form, and incorporating the entire sketch image, referred to as VAT.

As shown in the "White" row of Table 11, simply presenting groung truth-related region and reducing redundant information does not necessarily enhance model performance as shown in "Present All-GT". In contrast, the sketch method outperforms both "White" and surpasses the baseline in "Present All-GT" configuration. This suggests that while the sketch form benefits model performance, it does not achieve this simply by removing irrelevant information. Instead, it filters out redundancy while preserving essential semantic and structural details. Consistently, the results in Table 12 demonstrate that incorporating additional scaffolding information is beneficial. Retaining more relevant regions in the images increasingly boosts model performance. In this context, VAT functions similarly to selectively masking irrelevant details while retaining essential cues while retaining essential cues for reasoning.

# I  ANALYSIS ACROSS TASKS AND MODEL

## I.1  ANALYSIS ACROSS TASKS

While VAT consistently improves performance, the extent of performance gains from VAT varies across task types, as shown in Figure 6. VAT benefits visual reasoning through visual abstracts that stress implicit structural and semantic cues, stimulating abstract thinking from the visual side. It is particularly helpful for coarse-grained tasks, such as object relation reasoning (Odd-one-out, BLINK-Sem-corr, BLINK-Vis-corr) and spatial reasoning (BLINK-Spatial, CoSpace-Dir-rec, CoSpace-Rot-ang, CoSpace-Rot-dif), where VAT can effectively assists in filtering out irrelevant background clutters. However, in comparison to these tasks, VAT sometimes provides obscure details via visual abstract, limiting the performance gain for some object-centric perception tasks requiring fine-grained perception, such as CoSpace-count, BLINK-count, and HallusionBench-Illusion.

For tasks mainly focusing on perception, such as counting in Blink, the performance achieved with VAT surpasses that of CoT, with a 7.17% improvement in GPT-5 and a 1.44% improvement in Gemini 2.5-Flash. In tasks where reasoning also plays a significant role, CoT aids models in generating rationales, and the performance gap between VAT and CoT becomes marginal.

## I.2  GENERALITY ACROSS MODELS

We visualize the performance trends of applying VAT to GPT-4o, Gemini-2.0 Pro, and Qwen2.5-VL-32B in Figure 9. In conclude, while all models benefit from VAT, the extent of improvement varies considerably. GPT-4o exhibits the most substantial gain, achieving 15.33% over its CoT baseline, followed by Gemini-2.0-Pro (5.51%) and Qwen2.5-VL-32B (2.03%). This disparity may stem from the large model scale, extensive training data, and dedicated training recipe of GPT-4o, which allows it to better adapt to the abstract thinking pattern granted by sketches and align more closely with human cognitive reasoning. In contrast, Qwen2.5-VL-32B, with its relatively smaller scale, struggles to accommodate this shift and benefits less from VAT compared to GPT-4o or Gemini-2.0 Pro.

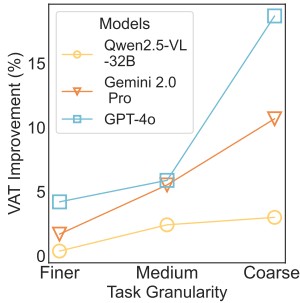

Figure 9: Visualization of influence of VAT across models and task granularity.

# J  ANALYSIS ON COMBINATION OF ABSTRACTION STYLES

Intuitively, features and information vary across different visual abstract styles. To further investigate the potential benefits of incorporating multiple abstract types, we conducted an experiment to simultaneously providing multiple distinct styles in Table 5. The concurrent input of three types of visual abstracts results in an average performance gain of +3.89% over the "Img-Only" baseline. It

Table 13: Experimental results (using GPT-4o) of simultaneously combining different visual styles. 2-Style Combo: Binary + OpenSketch; 3-Style Combo A: Binary + PhotoSketch + Opensketch; 3-Style Combo B: Aminie + Contour + Opensketch. VAT (Rule-based): rule-based selection of only one style according to their preferred tasks as discussed in Section 5.1.

| Method | Odd-One-Out | Visual Illusion | BLINK | | | | CoSpace | | | | | Avg. ACC | Avg. Gain |
|---|---|---|---|---|---|---|---|---|---|---|---|---|---|
| | | | Spatial | Count | Sem.Corr. | Vis.Corr. | Dir-Rec | Dir-Obj | Rot-Ang | Rot-Diff | Count | | |
| Img-Only | 25.49 | 55.56 | 82.52 | 65.83 | 53.96 | 86.05 | 40.40 | 46.00 | 58.33 | 50.50 | 40.00 | 54.97 | - |
| VAT (Ours) | 45.10 | 59.03 | 85.31 | 73.33 | 63.31 | 93.60 | 44.00 | 53.60 | 50.33 | 94.50 | 38.50 | 63.69 | +8.72 |
| VAT (Ruled-based) | 45.10 OpenSketch | 61.87 PhotoSketch | 87.41 PhotoSketch | 73.33 OpenSketch | 63.31 OpenSketch | 93.60 OpenSketch | 45.60 Contour | 55.20 Anime | 51.00 PhotoSketch | 94.50 OpenSketch | 41.00 PhotoSketch | 64.72 | **+9.75** |
| 2-Style Combo | 45.10 | 59.03 | 86.71 | 70.00 | 58.99 | 92.44 | 46.00 | 53.20 | 50.33 | 75.00 | 35.50 | 61.12 | +6.15 |
| 3-Style Combo A | 39.22 | 59.72 | 86.01 | 72.50 | 56.12 | 93.60 | 51.79 | 47.13 | 51.43 | 58.43 | 31.53 | 58.86 | +3.89 |
| 3-Style Combo B | 37.25 | 59.03 | 86.71 | 68.33 | 57.55 | 92.44 | 44.00 | 52.00 | 50.00 | 74.00 | 36.00 | 59.76 | +4.79 |

suggests that different forms of visual abstraction can contribute additional information from various perspectives, thereby enriching the reasoning process compared to the baseline.

However, it is noteworthy that the "Multi-style" approach underperforms "VAT (Best)", which achieves an average gain of +9.75%. This may be attributed to the fact that combining multiple visual abstract representations can overwhelm the model, impeding it to focus on the most relevant information for a given task. VAT is motivated to enhances reasoning by refining critical elements within a constrained amount of information. In contrast, incorporating multiple abstract styles increases the overall volume of information, which, although beneficial relative to baseline configurations, may dilute the model to take advantage of the most relevant aspects of the input.

In conclusion, though combining multiple styles of visual abstract yields a modest performance gain, it does not surpass the original VAT setting, suggesting that excessive messages can overwhelm the model and hinder its focus on the most relevant information.

# K  CASE STUDIES

To further illustrate the performance of VAT, we provide some typical cases in Figure 10. By presenting both the original image and its sketch, the model extracts key structural and semantic features such as line length or smoking area indicators and aligns them with the information from the original image for a more thorough understanding.

## K.1  DISCUSSION OF FAILURE CASES

We observe that VAT improves the overall performance across different task categories though, it degrades the model performance on specific tasks regarding different models. These failures can be categorized into two types: (1) VAT does not effectively reduce visual redundancy. This occurs when the original image is already visually simple or contains minimal noise, leaving limited room for simplification through sketch conversion, and thus offering negligible performance gain; (2) Conversely, VAT may remove critical visual details that are essential for accurate comprehension. In such cases, the MLLM may misidentify objects or miss important contextual information, resulting in wrong responses. Figure 11 exhibits some success and failure cases of VAT.

# L  FURTHER DISCUSSION COMBINING COT AND VAT

Table 14: Performance comparison across various task categories using GPT-4o with standard prompts, and enhanced with Chain-of-Thought (CoT), Visual Abstract Thinking (VAT), and their combination (CoT+VAT). The tasks are grouped into three categories: (1) those requiring visual perception and reasoning, (2) those focused on verbal knowledge and planning, and (3) tasks involving fine-grained visual detail.

| Method | Tasks focus more on **visual** perception and reasoning | | | | | | | | **Verbal** knowledge & planning tasks | | | | Fine-grained tasks | |
|---|---|---|---|---|---|---|---|---|---|---|---|---|---|---|
| | Visual Illusion | MME-Count | BLINK-Spatial | BLINK-Count | BLINK-Vis.Corr. | CoSpace-Dir-Rec | CoSpace-Dir-Obj | CoSpace-Rot-Diff | Odd-One-Out | MME-Comm. | CoSpace-Rot-Ang | CoSpace-PLA-Dec | BLINK-Sem.Corr. | CoSpace-Count |
| GPT-4o | 55.56 | 123.33 | 82.52 | 65.83 | 86.05 | 40.40 | 46.00 | 50.50 | 25.49 | 127.15 | **58.33** | 54.46 | 53.96 | 40.00 |
| +CoT | 54.17 | 158.33 | 84.62 | 72.50 | 88.95 | 43.55 | 43.32 | 77.84 | 29.41 | **176.38** | 57.19 | 63.38 | **66.19** | **47.00** |
| +VAT | **59.03** | **185.00** | **85.31** | 73.33 | **93.60** | **44.00** | **53.60** | **94.50** | 45.10 | 171.43 | 50.33 | 50.23 | 63.31 | 38.50 |
| +VAT & CoT | 58.33 | **185.00** | 85.11 | 69.17 | 88.95 | 36.00 | 52.23 | 68.50 | **47.06** | 174.89 | 55.00 | **72.41** | 58.99 | 37.56 |

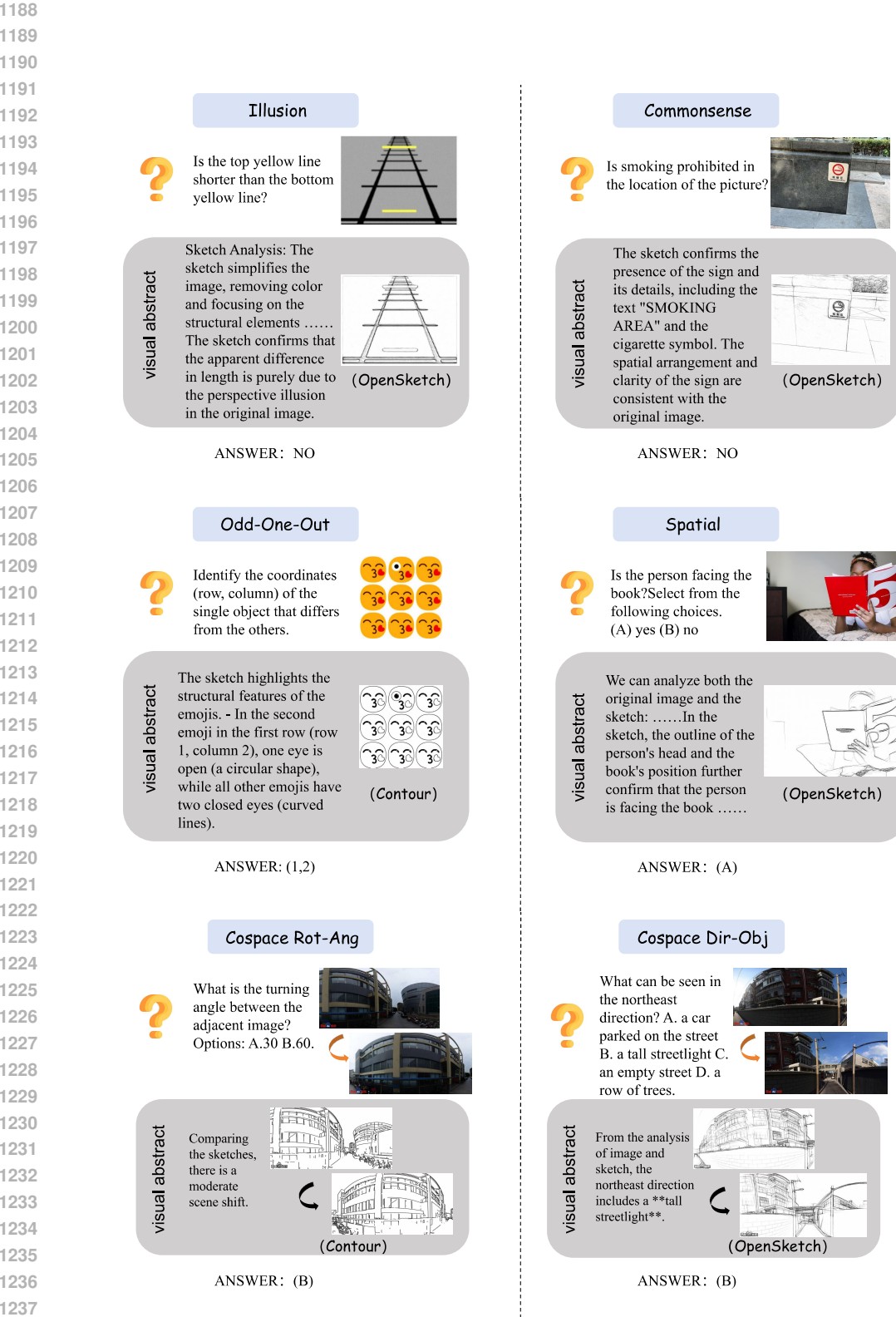

Figure 10: Examples of VAT. Each case pairs an original image with its visual abstract, demonstrating how the model leverages simplified yet semantically meaningful visual abstracts to enhance reasoning.

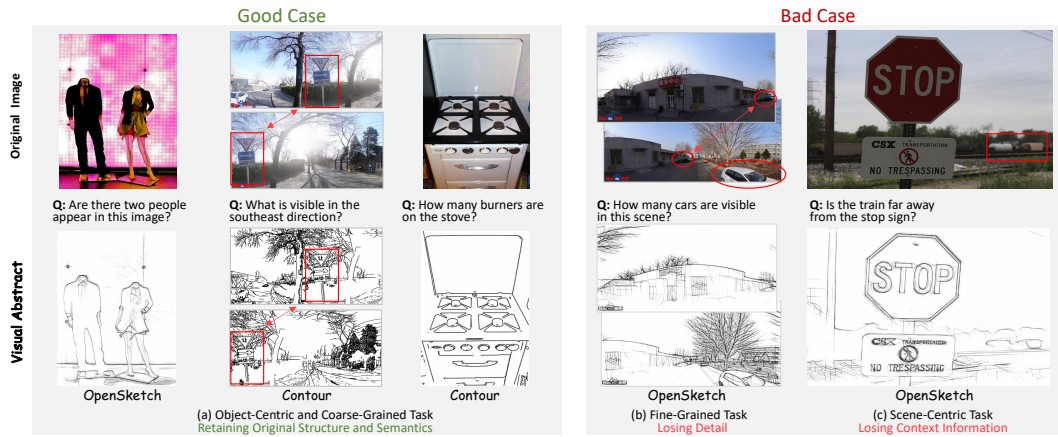

Figure 11: Success and failure cases of VAT. Converted visual abstracts retain essential relational and structural information, leading to successful improvements. While these visual abstracts sometime omits find-grained details and context as shown in the badcases, resulting in incorrect answers.

Table 14 provides a detailed analysis of the performance improvements resulting from the introduction of CoT, VAT, and their combination (+ VAT & CoT). The results show that although both CoT and VAT yield notable improvements across tasks compared with the standard prompts, the types of tasks they enhance differ significantly. We classify these tasks into three types: 1) tasks that focus more on visual perception and visual reasoning, 2) tasks that rely more on world knowledge and planning ability, and 3) tasks require fine-grained visual details. Our VAT demonstrates greater effectiveness in visually grounded tasks (the first type in Table 14), particularly those involving spatial understanding and directional inference. In contrast, CoT, which emphasizes explicit verbal reasoning, leads to substantial gains in tasks depend more on world knowledge where verbal reasoning and planning is especially effective. Consequently, these tasks can be further boosted by combining VAT with CoT, such as odd-one-out, and commonsense reasoning in MME. Notably, for the PLA-Dec task in the CoSpace benchmark, which requires both visual and verbal reasoning, the combined approach of CoT and VAT outperforms either method alone. Finally for tasks requiring fine-grained observations, VAT potentially omits some detailed visual context, and CoT achieves the best results instead.

## M    ABLATION STUDY ON TYPES OF VISUAL ABSTRACT

Table 15: Ablation study on the type of visual abstractions, including direct extraction and converting into sketches. Results are obtained using GPT-4o. Gain refers to the average relative improved percentage compared with the "None" baseline.

| Capabilities | Benchmark-*Task* | None | Direct Extraction | | Freehand Sketch Conversion | | | | Gain% |
|---|---|---|---|---|---|---|---|---|---|
| | | | Canny | Binary | PhotoSketch | Contour | Anime | OpenSketch | |
| Object -Centric Perception | BLINK - *Count* | 65.83 | 70.00 | 70.83 | 66.67 | 69.17 | 72.50 | **73.33** | 6.97% |
| | HallusionBench - *Illusion* | 55.56 | **61.87** | 61.11 | 61.11 | 59.72 | 59.03 | 59.03 | 8.55% |
| | CoSpace - *Dir-obj* | 46.00 | **55.60** | 55.20 | 53.60 | 55.20 | 55.20 | 53.60 | 18.99% |
| Object Relation Reasoning | Ours - *Odd-one-out* | 25.49 | 39.22 | 43.14 | 43.14 | 41.18 | 35.29 | **45.10** | 61.55% |
| | BLINK - *Sem. corr* | 53.96 | 57.89 | 57.97 | 53.24 | 58.27 | **63.31** | 63.31 | 8.00% |
| | BLINK - *Vis. corr* | 86.05 | 90.12 | 90.70 | 91.28 | 91.86 | 93.02 | **93.60** | 6.64% |
| Spatial Reasoning | BLINK - *Spatial* | 82.52 | 87.41 | **87.41** | **87.41** | 86.01 | 85.31 | 85.31 | 4.79% |
| | CoSpace - *Dir-rec* | 40.60 | 42.00 | 44.40 | 44.40 | **45.60** | 44.40 | 44.40 | 8.87% |
| | CoSpace - *Rot-ang* | 58.33 | 50.67 | 51.00 | 51.00 | 50.33 | 50.33 | 50.33 | -13.24% |
| | CoSpace - *Rot-dif* | 50.50 | 92.00 | 94.00 | 93.50 | 94.00 | 94.00 | **94.50** | 85.48% |
| Avg. Improve. of Visual Abstract Types | | - | +8.19 | +9.09 | +8.05 | +8.65 | +8.32 | **+9.77** | 8.87% |

In this section we test various types of visual abstractions and summaries corresponding trends presented by VAT. We conduct ablation study with GPT-4o to investigate the influence of six different types and styles of visual abstract, including canny edge detection Canny (1986) and image binarization, and deep-learning-based image-to-sketch models, including PhotoSketch Li et al. (2019), as well as Contour, Anime, and OpenSketch styles Chan et al. (2022). Experimental results regarding different model capabilities are reported in Table 15. We first conclude that OpenSketch style visual

abstract achieves the highest improvement on average for GPT-4o. This style of visual abstract largely preserve geometric details (depth and shape), rich semantic information, and artistic representations. This is followed by PhotoSketch, which retains boundary-like drawings that capture the outlines of the visual scene, including object boundaries, salient inner edges such as occluding contours, and prominent background edges.

# N  ATTENTION VISUALIZATION

We provide cases that visualize of the attention weights of Qwen-2.5-VL-32B when generating answer tokens in Figure 12, Figure 13 and Figure 14. In the ActiView case in Figure 12, it show that with the original image (upper), the model focuses more on objects inside the garage. While the visual abstract (lower) filters out most of the content inside the garage, and switches the focus to the badge on the wall, which successfully lead the model to this clue and correctly yields the answer "New South Wales". In the BLINK case in Figure 13, the correct answer to the question is "0" as no people are standing there. However, there is a black-shaded shape of a human body near the wall in the original image, which easily distracts the model. While in the visual abstract (lower), background texture are reorganized, further mitigating this distraction of attention. As a results, although there is a similar shape in the original image, the attention shifts to be more evenly distributed across the image via the visual abstract.

In the Odd-One-Out case shown in Figure 14, the question requires identifying the single object that differs from the others. On the original image, the attention distribution is scattered and lacks clear discriminative focus. In contrast, on the visual abstract, the model attends to the ground-truth target at position (2,2), as this egg contains a noticeably larger number of internal dots. Since the visual abstract preserves essential contour and geometric information, the model can more easily capture the anomaly. This demonstrates that the visual abstract effectively guides attention using semantic and geometric cues, leading to a more effective reasoning process.

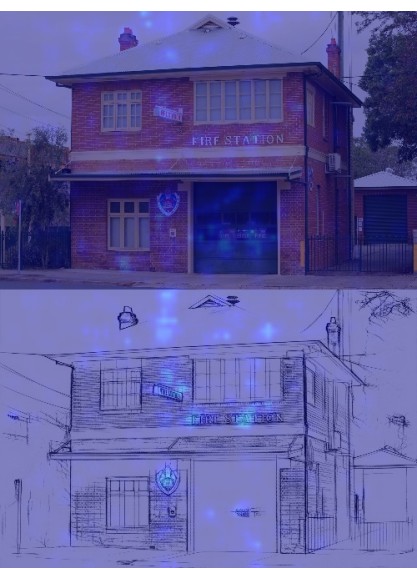 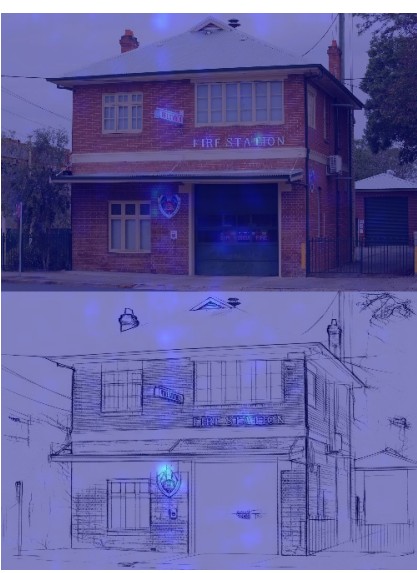

(a) Attention when generating token "Answer"          (b) Attention when generating token "Wales"

Figure 12: Visualization of attention shift of Qwen-2.5-VL-32B between the original image and visual abstract when generating answer tokens. The Question is *"Where is this photo taken?"*. The candidate options include: London, Paris, New South Wales, and New York. Image taken from ActiView benchmark. While the original image (upper) attracts most attention around the garage region, visual abstract (lower) refines this by filtering the corresponding context which consequently lead attention to shift to other more essential areas.

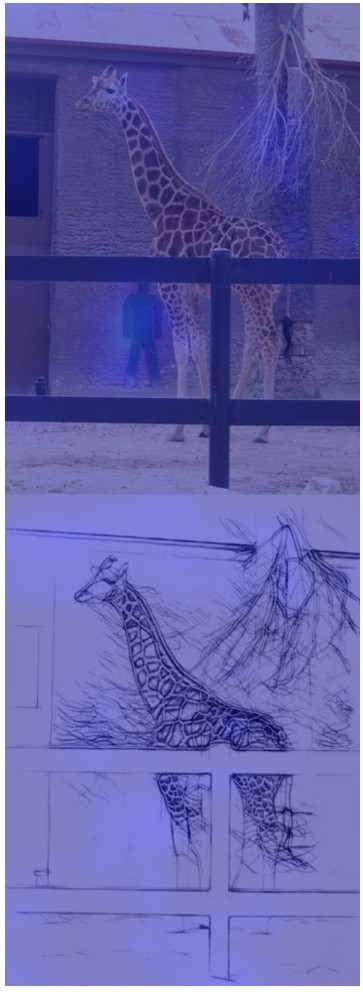

(a) Attention when generating token "Answer"   (b) Attention when generating token "0"

Figure 13: Visualization of attention shift of Qwen-2.5-VL between the original image and visual abstract when generating answer tokens. The Question is "*How many people are standing there?*". The candidate options include: 0, 1, 2 and 3. Image taken from BLINK benchmark. There is a human figure shape in the original image (upper). It is actually **a logo within the texture of wall, not a real human**, and it easily distracts the model. While in the visual abstract (lower), background texture are omitted, further refining this distraction of attention. The attention shifts to be more evenly distributed across the image.

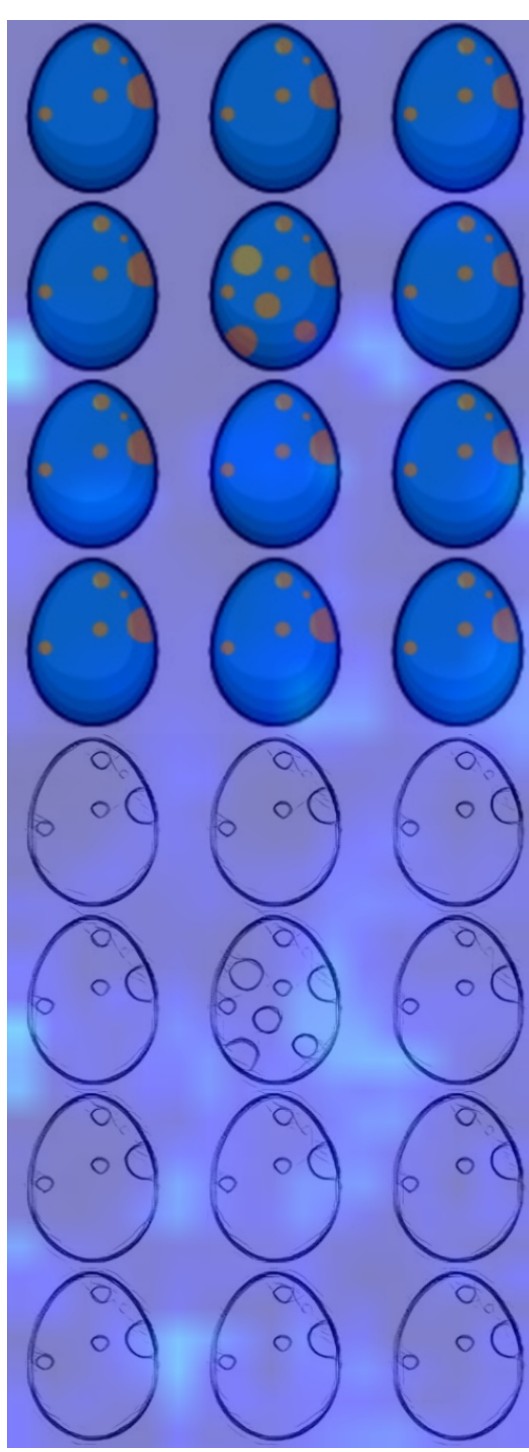

Figure 14: Visualization of the attention shift of Qwen-2.5-VL-32B between the original image and the visual abstract. The question is: "Identify the single object that differs from the others." The image is taken from the Odd-One-Out dataset. On the sketch image, the model correctly attends to the ground-truth target at position (2, 2), as this egg contains a larger number of internal dots. Since the sketch preserves the key contours and geometric structure of the eggs, the model can easily distinguish the anomalous one. The attention is therefore guided by both semantic and geometric information provided by the visual abstract. In contrast, on the original image, the attention distribution is scattered and lacks clear semantic relevance.

# O    TRAINING WITH VISUAL ABSTRACT THINKING

To verify that VAT is a generalizable reasoning paradigm that can be acquired not only through training-free prompting but also internalized via training, we post-trained the Qwen-2.5-VL-3B model using Group Relative Policy Optimization (GRPO), an efficient reinforcement learning algorithm. We collected 6,000 trajectories from Android Control, a GUI benchmark, to ensure that the model learns the underlying Visual Abstract Thinking pattern rather than merely overfitting to the evaluated benchmarks.

As shown in Table 16, the RL-trained model achieves significant performance improvements across multiple tasks, with an average gain of +4.52%. These results confirm that "Thinking with Visual Abstract" is a learnable reasoning paradigm.

Table 16: Performance comparison between Qwen-2.5-VL-3B and VAT-3B. VAT-3B is derived by fine-tuning Qwen-2.5-VL-3B using Group Relative Policy Optimization (GRPO) on 6,000 trajectories from the Android Control dataset.

| Method | Odd-One -Out | Visual Illusion | Acti View | Text VQA-GT | BLINK | | | | Avg. Gain |
|---|---|---|---|---|---|---|---|---|---|
| | | | | | Spatial | Count | Sem.Corr. | Vis.Corr. | |
| Qwen-2.5-VL-3B | 0.00 | 57.64 | **69.38** | 55.32 | 78.32 | 60.83 | **30.94** | 37.21 | – |
| **VAT-3B** | **21.57** | **61.81** | 59.69 | **69.43** | **83.92** | **61.67** | 28.78 | **38.95** | **+4.52** |

