# OpenReview forum: "Visual Abstract Thinking: Enhancing Multimodal Reasoning via Visual Abstraction"
_ICLR.cc/2026/Conference — Submitted to ICLR 2026_

### Official Review · Reviewer_hRp8 · 2025-10-18

**Soundness:** 3
**Presentation:** 3
**Contribution:** 3
**Rating:** 6
**Confidence:** 4

**Summary:**

This paper proposes Visual Abstract Thinking (VAT), a novel reasoning paradigm for multimodal large language models (MLLMs). Instead of relying on explicit textual reasoning such as Chain-of-Thought (CoT) or tool-based intermediate steps, VAT introduces visual abstracts that highlight essential visual elements while suppressing redundant information. Experiments demonstrate that VAT consistently improves visual reasoning performance across different model scales and reasoning modes, with particularly strong gains on spatial and relational reasoning tasks.

**Strengths:**

The paper presents a thorough experimental study across multiple models, reasoning paradigms, and task types. The ablation analyses are detailed and well-structured, offering meaningful insights.
The overall narrative is well-organized and easy to follow.
And the reported results demonstrate that VAT achieves both higher accuracy and lower computational cost.

**Weaknesses:**

While the results are strong overall, the paper would benefit from a more explicit discussion of failure cases or negative results. For instance, Table 1 indicates certain tasks where VAT leads to performance drops. A deeper analysis of why abstraction harms those tasks  would make the work more balanced and informative.

**Questions:**

1. Beyond the benchmarks used in the paper, I am curious how VAT would perform on tasks that are already abstract or sketch-like in nature—for example, geometry reasoning or diagram understanding tasks where the input is sparse and symbolic rather than visually detailed. Since such tasks already emphasize structural abstraction, would VAT still provide benefits, or could it potentially remove critical geometric cues and thus degrade reasoning?

2. In Table 1, some tasks show performance drops under VAT while CoT yields gains. Combined with the prompt formulation in Appendix B.1, where the instruction enforces “you must fully utilize the provided visual abstracts,” I wonder if a more flexible prompting scheme could help. Specifically, have the authors considered allowing the model to autonomously decide whether or not to use the visual abstract (e.g., “you may use the visual abstract if needed”) and then analyze the resulting task-wise performance differences?

---

> ### Author Response · Authors · 2025-11-25
> **Response to Reviewer hRp8 (1 of 2)**
>
> Thank you for your careful reading and for recognizing our thorough experimental study and detailed ablation analyses, as well as the novelty of the VAT reasoning paradigm. We address your concerns and include further discussion regarding your questions as belows:
>
> **1. Discussion on Failure Cases and Performance Drops (Weakness 1)**
>
>   We provided failure cases and corresponding analysis in Appendix J. Figure 11 visualise two typical failure cases where visual abstraction causes the model to overlook critical details or contextual cues. Although being significantly effective, VAT still suffers from inadvertent losing of minor details and context required for reasoning. We summarise two potential causes of such performance drops: (1) strong language prior weakens the benefit of visual abstract thinking, and (2) the increased visual token load that exceeds the capacity of smaller models, as follows:
>
>   - **Many existing MLLMs still rely heavily on linguistic priors.**
>
>     Many MLLMs are built upon well-tuned language models. They do NOT consistently integrate visual token during reasoning [1]. **This imbalance can reduce the benefit of visual abstraction in tasks that require thinking from the visual side**. Larger models potentially achieve a better balance between visual and textual processing. Their stronger gains with VAT, such as the improvements on GPT 5(+2.21) and Gemini-2.5-Flash(+3.65) compared with the smaller Qwen-2.5-VL-32B(+0.94), support this interpretation.
>
>   - **Increased visual tokens challange the multimodal comprehension ability of models**
>
>     VAT doubles the visual token input by maintaining both the original image and its abstract. It is affordable for tasks with a few images. While for multi-image tasks like CoSpace (4 images/query), VAT increases the input to 8 images. Moreover, these images are HD images, which further challanges the capability of models. It requires models to understand the relation and capture the dependency among the image sequence.
>
> **2. Does VAT still provide benefits to already abstract or sketch-like images?（Question 1）**
>
>   VAT is designed to retain core structural and relational patterns, making it effective even for highly abstract inputs.
>
>   Our evaluation has included tasks with already abstracted images such as **Odd-One-Out** and **Visual Illusion** (results reported in Table 1). These tasks rely on shapes and structures rather than natural scenes. As cases visualized ** in Figure 10**, VAT preserves key geometric relationships while filtering noise without degrading the reasoning performance. Results on these tasks show significant gains, validating that VAT can still provide benefits to those images with highly concentrated visual information.
>
>
> [1] Liu, C., et al. (2025). More Thinking, Less Seeing? Assessing Amplified Hallucination in Multimodal Reasoning Models. NeurIPS 2025.

---

> > ### Author Response · Authors · 2025-11-25
> > **Response to Reviewer hRp8 (2 of 2)**
> >
> > **3. Impact of flexible prompting scheme (Question 2)**
> >
> >   Thanks for this insightful suggestion. We additionally implement the more flexible instruction (“You may use the visual abstract if needed”) in response to this question. Moreover, to further investigate the influence of tone of verbal instruction, we also implement a version without literal instruction (omitting the "you may/must use" statement).
> >
> >   - For completeness, the flexible prompt used in this experiment is shown below:
> >     ```
> >     Each image will be provided together with a corresponding sketch, which is directly converted from the original image. The sketch helps you determine essential components in the image, including but not limited to spatial, structural, relational, and conceptual features, which can assist in your reasoning process.You should use both the original image and the sketch to inform your reasoning process. You may use the visual abstract if needed.
> >     ```
> >
> >     The corresponding results are shown in the "VAT *(Prompt: "you may use")*" lines, compared with results of our strict prompt (“you must fully utilize the provided visual abstracts”) shown in "VAT" lines. "VAT *(No Prompt)*" refers to version without literal instruction.
> >
> >   - Results of this experiment are as follows:
> >
> > | Method | Odd-One -Out | Visual Illusion | Acti View | Text VQA-GT | blink-spatial | blink-counting | blink-sem_corr | blink-vis_corr | CoSpace-dir-rec | CoSpace-dir-obj | CoSpace-rot-ang | CoSpace-rot-diff | CoSpace-counting | Avg. Gain |
> > | :--- | :---: | :---: | :---: | :---: | :---: | :---: | :---: | :---: | :---: | :---: | :---: | :---: | :---: | :---: |
> > | **Gemini-2.5-Flash** | 66.67 | 56.94 | 77.13 | 76.18 | 86.71 | 69.17 | 57.55 | 83.14 | 49.60 | 52.82 | 61.33 | 78.50 | 38.69 | - |
> > | +VAT | 72.55 | 59.03 | 75.58 | 76.40 | 84.82 | 66.10 | 56.12 | 93.02 | 52.21 | 60.24 | 75.59 | 86.87 | 43.50 | +3.65 |
> > | +VAT *(Prompt: "you may use")* | 74.51 | 56.94 | 77.13 | 76.94 | 85.31 | 68.33 | 57.55 | 90.12 | 46.80 | 58.00 | 81.33 | 89.50 | 41.50 | +3.81 |
> > | +VAT *(No Prompt)* | 72.55 | 56.25 | 76.36 | 75.92 | 86.01 | 68.33 | 57.55 | 92.42 | 44.94 | 50.00 | 82.19 | 89.00 | 37.59 | -1.14 |
> > | **GPT-5** | 70.59 | 59.03 | 87.20 | 82.10 | 86.71 | 71.67 | 69.78 | 94.08 | 68.00 | 72.40 | 65.00 | 78.50 | 45.23 | - |
> > | +VAT | 83.67 | 61.81 | 80.47 | 81.50 | 88.03 | 75.83 | 71.94 | 95.86 | 68.00 | 77.60 | 60.33 | 87.00 | 47.00 | +2.21 |
> > | +VAT *(Prompt: "you may use")* | 76.47 | 65.97 | 80.23 | 81.53 | 86.71 | 73.33 | 66.91 | 95.35 | 68.00 | 74.40 | 62.33 | 87.00 | 46.50 | +0.96 |
> > | +VAT *(No Prompt)* | 78.43 | 59.72 | 77.91 | 81.65 | 84.62 | 74.17 | 65.47 | 95.35 | 66.40 | 77.60 | 61.33 | 87.50 | 45.00 | +0.37 |
> > | **Qwen-2.5-VL-32B** | 37.25 | 60.42 | 71.76 | 69.65 | 83.92 | 74.17 | 47.48 | 77.91 | 36.33 | 48.12 | 55.56 | 68.84 | 34.54 | - |
> > | +VAT | 35.29 | 62.50 | 68.75 | 77.00 | 84.67 | 74.17 | 51.08 | 82.56 | 31.82 | 56.80 | 57.33 | 64.80 | 31.41 | +0.94 |
> > | +VAT *(Prompt: "you may use")* | 31.37 | 59.03 | 74.81 | 76.91 | 86.71 | 69.17 | 55.40 | 79.65 | 32.40 | 53.20 | 55.00 | 65.00 | 32.00 | +0.36 |
> > | +VAT *(No Prompt)* | 35.29 | 60.42 | 76.74 | 78.41 | 88.11 | 69.17 | 51.80 | 83.14 | 31.20 | 51.60 | 54.00 | 62.00 | 35.00 | +0.84 |
> >
> >   - Conclusions from the experimental results:
> >
> >     - **Performance increases with the strictness of the instruction.**
> >       We observe that our strict prompting tone yields the highest performance in most models. Differently, an interesting exception appears on Gemini-2.5-Flash, which achieves its best results under the flexible prompt. A reasonable interpretation is that Gemini-2.5-Flash is better calibrated between language and vision, allowing it to invoke the abstract only when helpful rather than being explicitly forced to use it.
> >
> >     - **Strict instructions are generally necessary to prevent models from ignoring the visual abstract and reverting to language-only reasoning.**
> >
> >       The results indicate that strict prompting is important because it enforces a consistent “thinking with images” pattern. Without this constraint, many models revert to their strong internal linguistic priors or rely solely on the raw image, effectively ignoring the abstraction signal. The strict prompt compels integration of the visual abstract, ensuring that the structural cues provided by VAT are actually utilized.

---

> > > ### Comment · Reviewer_hRp8 · 2025-11-26
> > >
> > > Thanks for the detailed response. I have no further concerns.

---

### Official Review · Reviewer_mHuw · 2025-10-25

**Soundness:** 2
**Presentation:** 3
**Contribution:** 2
**Rating:** 4
**Confidence:** 4

**Summary:**

This paper proposes Visual Abstract Thinking (VAT), which generates an abstract representation of an original image and uses it together with the original image as input to an MLLM model. This approach allows the model to focus more on the essential visual elements, concepts, and structural features. The authors verify and discuss the effectiveness of VAT through experiments and analysis.

**Strengths:**

From the experimental results, VAT is shown to be helpful for certain types of multi-modal reasoning (coarse-grained tasks). I also appreciate the ablation analyses presented by the authors in session 4 and 5, including but not limited to the studies on the impact of providing different proportions of sketches, the influence of abstract style combinations and selections, as well as the efficiency.

**Weaknesses:**

1. Although the authors discuss the differences between Visual Tool-Using Agents and VAT, VAT can still be treated as a special tool-using case.The generation of the abstract representation of an original image relies on other deep learning models rather than being produced by the model itself. These deep learning models should therefore be considered as the model’s tools. This limits the novelty of the approach, making it more like providing the agent with a specialized tool and corresponding result. Considering that different types of tasks may benefit from different abstract styles, making the model to autonomously select and invoke the appropriate styles from different model (abstraction representation generation tool), could potentially yield even better results.

2. As the authors mention in Section 5.1, VAT is particularly helpful for coarse-grained tasks, but its performance gains are limited for the tasks that require fine-grained perception. This indicates that the VAT approach is not sufficiently general, and instead appears to be a task-specific design tailored for particular types of problems.

**Questions:**

1. What is the relationship between VAT and tool-using agents;

2. Do you consider researching ways for the model to autonomously generate visual abstraction representations to enhance its own visual reasoning ability, rather than relying on external tools.

---

> ### Author Response · Authors · 2025-11-25
> **Response to Reviewer mHuw (1 of 2)**
>
> We sincerely thank the reviewer for the constructive feedback and for recognizing the effectiveness of our approach in multimodal reasoning, as well as appreciating the depth of our ablation analyses regarding sketch proportions and efficiency.
>
> **1. Distinction between VAT and Tool-Using (Weakness 1 & Question 1)**
>
>   We argue that VAT represents a distinct **cognitive paradigm** rather than a standard tool-using case. The core contribution lies in identifying visual abstraction as an effective reasoning pathway. VAT reveals a reasoning pathway that can be externalized (via prompting) and also internalized (via training). Detailed clarifications are as follows:
>
>   - **VAT validates a cognition inspired "reasoning paradigm" rather than an engineering heuristic.**
>
>     VAT mimics efficient human cognition. It aligns with Marr’s Primal Sketch [1] (extracting intensity changes), Biederman’s RBC Theory [2] (prioritizing structural components), and Tversky’s Schematic Abstraction [3] (reducing cognition load via simplification). Our work validates that filtering redundancy to focus on structure is also effective for MLLMs. The novelty lies in establishing this cognitive pathway, independent of the implementation method (prompting or training).
>
>   - **VAT is a learnable reasoning paradigm that can be internalized via Reinforcement Learning (RL).**
>
>     Models can be trained to actively utilize visual abstracts based on context rather than relying solely on external pipelines. We validated this by fine-tuning Qwen-2.5-VL-3B using RL. As shown below, the model achieves significant gains (+4.52%), confirming that "Thinking with Visual Abstract" is a learnable capability. We will provide detailed training steps in our revision.
>
>     | Method | Odd-One-Out | Visual Illusion | Acti View | Text VQA-GT | blink-spatial | blink-counting | blink-sem_corr | blink-vis_corr | Avg. Gain |
>     | :--- | :---: | :---: | :---: | :---: | :---: | :---: | :---: | :---: | :---: |
>     | **Qwen-2.5-VL-3B (w/o training)** | 0.00 | 57.64 | 69.38 | 55.32 | 78.32 | 60.83 | 30.94 | 37.21 | - |
>     | **VAT-3B** | 21.57 | 61.81 | 59.69 | 69.43 | 83.92 | 61.67 | 28.78 | 38.95 | +4.52 |
>
>   - While for tool-using methods, external tool-calling are still required during inference. Even upon training, models better learn to use external tools, rather than acquiring internal thinking paradigm.
>
> **2. Autonomous selection of styles (Weakness 1)**
>
>   Thanks for raising the discussion regarding style selection. We address this concern in the following two aspects:
>
>   - **Style selection results**: We implement a rule-based selection according to the analysis of effects presented by different styles in Figure 6 in Section 5.1. The results of style selections and combinations are reported in Section 5.2. **These results already indicate that style selection do yield better results.** The analysis in Section 5.1 provide insights of how to conduct selection, and the Section 5.2 validates the effectiveness of selection, **encouraging and posing furture exploration of such mechanism.**
>
>   - **Discussion of further autonomous selection**: We agree that autonomously selecting styles is also beneficial. A possible way is to train via multi-turn Reinforcement Learning, controling the interaction visual abstract by special tokens. **Considering that further investigation of autonomous selection involves data preperation, training method exploration and analysis regarding the performance of different models, which could exceed the page limitation of a single paper**. We sincerely appreciate this suggestion and will place corresponding research in our future plan.
>
>
> [1] Marr, D. (2010). Vision: A computational investigation into the human representation and processing of visual information. MIT press.
>
> [2] Biederman, I. (1987). Recognition-by-components: a theory of human image understanding. Psychological review, 94(2), 115.
>
> [3] Tversky, B. (2013). Visualizing thought. In Handbook of human centric visualization (pp. 3-40). New York, NY: Springer New York.

---

> ### Author Response · Authors · 2025-11-25
> **Response to Reviewer mHuw (2 of 2)**
>
> **3. Generalizability of VAT on fine-grained tasks (Weakness 2)**
>
>   Although VAT is particular beneficial to coarse-grained tasks, this does not indicate that VAT is not helpful to fine-grained tasks. As we discussed in Section 5.1, we argue that VAT is NOT a narrow, task-specific design. The visual abstractions of VAT could also guide models to better notice and digest required visual information for fine-grained tasks. There are two primary reasons:
>
>   - **VAT implements a universal structural or geometric information rather than a task-specific heuristic.**
>
>     Multimodal reasoning inherently benefits from decomposing complex scenes. VAT mimics the biological "primal sketch" [1] to filter low-level noise, providing a clear structural layout.
>
>   - **The Dual-Input setting preserves fine-grained details.**
>
>     - VAT enhances fine-grained perception because the Dual-Input design does not replace the raw image. The visual abstraction provides a spatial prior to guide the understanding of salient regions, and the original fine-grained cues such as color and texture remain intact.
>
>     - This design leads to clear gains on fine-grained benchmarks. We conduct additional experiment on fine-grained tasks involving color features and salient observations such as MME-Color and V*(GPT-4V-Hard). On MME-Color, GPT-5 improves by **+20.00** (ACC) and **+8.34** (ACC+) over the baseline of 73.33 (ACC) and 88.33 (ACC+). For V*(GPT-4V-Hard), its performance rises by +11.77 (from 52.94 to 64.71). Although the improvements are not as remarkable as for coarse-grained tasks, these results still validate the effectiveness of VAT on fine-grained task.
>
>     _Note: ACC+ refers to the strict accuracy metric in which an instance is counted as correct only when both associated questions are answered correctly._
>
> **4. Autonomous Visual Abstract Generation (Question 2).**
>
>   - **Our primary research motivation is to examine whether MLLMs possess the inherent capability to utilize the Visual Abstract Thinking (VAT) paradigm** align with human cognition, and to systematically analyze how such paradigm influence multimodal reasoning performance. Upon understanding how this paradigm affects the reasoning behavior of MLLMs, we can better step forward towards research of autonomous generation.
>
>   - **While enabling the model to autonomously generate visual abstracts is an insightful and promising direction**, establishing such a generative method introduces significant complexity that far exceeds the scope and page limits of this paper. As unified models advance rapidly, we are actively exploring task-dependent generative strategies for producing visual abstracts. This exactly align with our follow-up work. Also, recent released models such as Gemini 3 demonstrate impressive unified capabilities in both image-conditioned generation and multimodal understanding, suggesting that high-quality, task-aware visual abstraction may soon become feasible.
>
>   We really appreciate this suggestion and are glad that our current research plan align with this suggestion. Besides, we will also consider the suggestion of autonomous selection mentioned in Weakness 1.
>
> [1] Marr, D. (2010). Vision: A computational investigation into the human representation and processing of visual information. MIT press.

---

### Official Review · Reviewer_8NEK · 2025-10-25

**Soundness:** 3
**Presentation:** 3
**Contribution:** 2
**Rating:** 4
**Confidence:** 5

**Summary:**

This paper introduces Visual Abstract Thinking (VAT), a new paradigm designed to improve how MLLMs reason about images. The authors argue that current methods, like CoT, rely on inefficient and distracting verbose text descriptions. VAT replaces this textual reasoning by instead prompting the model with a simplified "visual abstract" of the image, such as a sketch or contour map. This approach helps the MLLM focus on essential visual structures, leading to more accurate and efficient reasoning. The paper's main contributions are demonstrating that VAT achieves higher accuracy than standard CoT and GPT-5 baselines while simultaneously being more cost-effective, requiring significantly fewer tokens and less runtime.

**Strengths:**

1. The proposed approach is simple yet effective.
2. The method demonstrates broad applicability across various MLLMs, including both proprietary and open-source models.
3. The ablation study on VAT's underlying mechanisms is convincing and provides valuable insights.

**Weaknesses:**

1. This method relies on external image to sketch tool/models, which is neither context aware, nor task relevant.
2. The 'Visual Abstract Thinking' (VAT) nomenclature could be misleading. Given that the visual abstraction is provided as input rather than generated by the model, 'Visual Abstract Prompting' might be a more accurate descriptor.
3. The analysis lacks certain ablation studies and baseline comparisons:
- An ablation study is needed to show how different sketching methods affect performance.
- A comparison against strong tool-use baselines for perception-intensive tasks. (e.g., Set-of-Mark [1], thinking with images) are missing, considering VAT's advantage of concentrated perception.

Reference:

[1] Yang, Jianwei, et al. "Set-of-mark prompting unleashes extraordinary visual grounding in gpt-4v." arXiv preprint arXiv:2310.11441 (2023).

**Questions:**

1.Could the authors clarify the 'Rand-GT' condition in Figure 4? Its relationship to 'All-GT' is unclear from the figure.

2. There appear to be numerical inconsistencies in the reported gains in Table 1 (e.g., for Qwen2.5-VL+VAT on the ActiView and TextVQA-GT benchmarks). Please carefully verify all reported results.

---

> ### Author Response · Authors · 2025-11-25
> **Response to Reviewer 8NEK (1 of 2)**
>
> We sincerely appreciate your recognize on VAT as "simple yet effective", and your acknowledgement of the value of our ablation studies. We address your concerns in detail as follows:
>
> **1. VAT relies on external tools and is not context aware (Weakness 1).**
>
>   We propose VAT as a thinking mechanism to reveal visual thinking potential and provide detailed analysis, rather than introducing a external tool-using approach. We conduct additional experiments to show that this ability is learnable and internalized via training.
>
>   - **VAT is a learnable paradigm that can be internalized via Reinforcement Learning (RL).**
>
>     Models can be trained to actively understand visual abstracts based on context rather than relying solely on external pipelines. We validated this by fine-tuning Qwen-2.5-VL-3B using RL. As shown below, the model achieves significant gains (+4.52%), confirming that Visual Abstract Thinking is a learnable capability, rather than simply an external tool-using approach. We wil provide detailed training steps in our revision.
>
>     | Method | Odd-One-Out | Visual Illusion | Acti View | Text VQA-GT | blink-spatial | blink-counting | blink-sem_corr | blink-vis_corr | Avg. Gain |
>     | :--- | :---: | :---: | :---: | :---: | :---: | :---: | :---: | :---: | :---: |
>     | **Qwen-2.5-VL-3B (w/o training)** | 0.00 | 57.64 | 69.38 | 55.32 | 78.32 | 60.83 | 30.94 | 37.21 | - |
>     | **VAT-3B** | 21.57 | 61.81 | 59.69 | 69.43 | 83.92 | 61.67 | 28.78 | 38.95 | +4.52 |
>
>   - **We addressed task relevance via a rule-based selection strategy implemented in Section 5.2.**
>
>     To address the task-style relevance, we had implemented a rule-based selection strategy in Section 5.2. This is implemented according to the analysis of effects presented by different styles in Figure 6 in Section 5.1. These analysis reveal task relevance of style. To bridge the gap between context-agnostic conversion and specific reasoning needs, we further implement rule-based strategy upon these analysis to selects the optimal abstraction style corresponding to the task type, ensuring the most relevant structural features are highlighted. Results show that task-relevant selection slightly improves the average performance gain from +8.72 to +9.75.
>
> **2. Visual Abstract Thinking or Visual Abstract Prompting (Weakness 2).**
>
>   We appreciate the careful suggestion. However, we believe "Visual Abstract Thinking" (VAT) accurately defines our contribution as a cognitive reasoning paradigm rather than a mere prompting technique. Although it seems like a prompting technique, our motivation is beyond finding an effective approach to achieve high scores on benchmarks. We address that visual abstract thinking stems from cognitive theories and is also learnable by models as follows:
>
>   - **VAT defines a cognitive paradigm verifiable via both prompting and training.**
>
>     As demonstrated by our RL experiments (see response to Weakness 1), the VAT mechanism can be internalized as a learned policy. Whether implemented via inference-time prompting or training is secondary; the core contribution is the **establishment of "Visual Abstraction" as a valid and effective reasoning pathway for MLLMs.**
>
>   - **We validate that the human cognitive strategy of "schematic abstraction" is effective for MLLMs.**
>
>     Our work fundamentally mimics efficient human cognition rather than engineering heuristics. It aligns with David Marr’s theory of the "primal sketch" by extracting essential intensity changes [1], prioritizes structural "components" for recognition as proposed by Biederman [2], and utilizes schematic abstraction to reduce cognitive load as described by Tversky [3]. By forcing the model to process information through this structural information, VAT constitutes a distinct "thinking" pattern that lowers cognitive consumption.
>
>   - VAT is interalizable for models, as we validated in the response to Weakness 1. The results align with our motivation that **VAT is not simply a prompting method, but a learnable thinking paradigm that can be applied without prompting.**
>
> [1] Marr, D. (2010). Vision: A computational investigation into the human representation and processing of visual information. MIT press.
>
> [2] Biederman, I. (1987). Recognition-by-components: a theory of human image understanding. Psychological review, 94(2), 115.
>
> [3] Tversky, B. (2013). Visualizing thought. In Handbook of human centric visualization (pp. 3-40). New York, NY: Springer New York.

---

> ### Author Response · Authors · 2025-11-25
> **Response to Reviewer 8NEK (2 of 2)**
>
> **3. Ablations of different sketch and Baselines of Set-of-Mark (SoM) (Weakness 3).**
>
>   - Ablations of different sketch：
>     - **Different sketching methods lead to predictable and modest performance differences, and Figure 6 provides a clear summary of these effects.** Our ablations show that pixel-based sketches such as Canny and Binary provide slightly stronger gains on object-centric tasks by filtering background clutter. OpenSketch offers the largest improvement on object relation reasoning because it preserves structural cues. Spatial reasoning shows only small variation across sketch types. We will include a detailed breakdown of results contribute to Figure 6 in the Appendix in our revision.
>
>   - Comparison with different baselines：
>     - In Table 2 in our paper, **we demonstrate that VAT outperforms strong tool-using baseline, Visual SketchPad, while being significantly more cost-effective**. Visual SketchPad relies on complex pipelines involving heavy external models (e.g., Grounding DINO, SAM) and code execution, incurring high latency. VAT achieves superior accuracy through efficient structural guidance without heavy-weight tool invocation.
>     - We conducted an additional experiment in which the visual abstract was replaced with recommended Set-of-Mark (SoM) images, as shown in the table below. Results indicate SoM underperforms VAT. While SoM excels at symbolic grounding (discrete markers), **VAT provides a holistic structural view that filters redundancy. This allows the model to focus on essential geometric relationships required for high-level reasoning.**
>
> | Method | Odd-One -Out | Visual Illusion | Acti View | Text VQA-GT | blink-spatial | blink-counting | blink-sem_corr | blink-vis_corr | CoSpace-dir-rec | CoSpace-dir-obj | CoSpace-rot-ang | CoSpace-rot-diff | CoSpace-counting | Avg. Gain |
> | :--- | :---: | :---: | :---: | :---: | :---: | :---: | :---: | :---: | :---: | :---: | :---: | :---: | :---: | :---: |
> | **Gemini-2.5-Flash** | 66.67 | 56.94 | 77.13 | 76.18 | 86.71 | 69.17 | 57.55 | 83.14 | 49.60 | 52.82 | 61.33 | 78.50 | 38.69 | -|
> | +VAT | 72.55 | 59.03 | 75.58 | 76.40 | 84.62 | 66.10 | 56.12 | 93.02 | 52.21 | 60.24 | 75.59 | 86.87 | 43.50 | **+3.65** |
> | +SOM | 62.75 | 59.72 | 74.81 | 77.69 | 83.92 | 72.50 | 58.99 | 90.70 | 48.40 | 52.00 | 69.33 | 87.50 | 39.50 | +1.80 |
> | **GPT-5** | 70.59 | 59.03 | 67.20 | 82.10 | 86.71 | 71.67 | 69.78 | 94.08 | 68.00 | 72.40 | 65.00 | 78.50 | 45.23 | -|
> | +VAT | 83.87 | 61.81 | 80.47 | 81.50 | 88.03 | 75.83 | 71.94 | 95.96 | 68.00 | 77.60 | 60.33 | 87.00 | 47.00 | **+2.21** |
> | +SOM | 70.59 | 61.11 | 79.07 | 81.94 | 85.31 | 68.33 | 68.35 | 95.93 | 66.40 | 73.20 | 63.00 | 83.00 | 49.00 | -0.39 |
> | **Qwen-2.5-VL-32B** | 37.25 | 60.42 | 71.76 | 69.65 | 83.92 | 74.17 | 47.48 | 77.91 | 36.33 | 48.12 | 55.56 | 68.84 | 34.54 | -|
> | +VAT | 35.29 | 62.50 | 68.75 | 77.00 | 84.67 | 74.17 | 51.06 | 82.56 | 31.82 | 56.80 | 57.33 | 64.60 | 31.41 | **+0.94** |
> | +SOM | 39.22 | 61.11 | 74.42 | 78.98 | 82.52 | 68.33 | 48.20 | 82.56 | 32.00 | 48.80 | 57.67 | 63.00 | 29.50 | -0.13 |
>
> **4. Clarification on 'Rand-GT'(Question 1)**
>
>   In Figure 4, 'Rand-GT' refers to a condition where the visual abstract randomly includes only a **subset of the Ground Truth (GT) regions**, mixed with noise. This contrasts with 'All-GT', which presents all essential regions. This comparison verifies that **performance scales positively with the completeness of the provided structural cues.**
>
> **5. Numerical Inconsistencies (Question 2)**
>
>   We sincerely apologize for this. Upon a thorough verification of our experimental logs, we identified a data transcription error where two specific values for Qwen2.5-VL+VAT (on ActiView and TextVQA-GT) were inadvertently misplaced. We have corrected these in revision (highlighted in orange). We have verified that these are isolated clerical errors and **do not alter the overall performance trends or our main conclusions.**

---

> > ### Comment · Reviewer_8NEK · 2025-11-26
> >
> > Thank you for the explanations and the newly added ablation study and baselines. I believe these additions have successfully cleared my concerns regarding Weakness 3 and the associated questions. However, the core concerns of **Weakness 1** and **Weakness 2** still remain.
> >
> > ### **W1: Visual Abstract Representation**
> >
> > My primary concern is that the **visual abstract representation** extracted from the original image lacks crucial properties: it is neither **context-aware** nor **task-relevant**.
> >
> > While I do not contest the MLLMs' ability to understand and reason with the visual abstracts conditioned on the provided context, the representation itself is deficient. Specifically, for a representation to be truly **task-relevant**, the key is to **disambiguate task-relevant objects from task-irrelevant objects**, a crucial mechanism demonstrated in work such as RT-sketch [1]. Simply adopting a **sketch style** does not inherently achieve this necessary level of task-relevant abstraction.
> >
> > ### **W2: Ambiguity of "Visual Abstract Thinking"**
> >
> > The claim "Visual abstract thinking" remains **ambiguous** in the context of this work.
> >
> > For me, this term denotes the capability to form a visual representation that is simultaneously **abstract** and **task-relevant**. I acknowledge and agree with the clarification provided by the authors to reviewer mHuw, suggesting the term **"Thinking with Visual Abstract"** better aligns with the specific problem setting in this work—the model's ability to reason using pre-extracted visual abstractions.
> >
> > However, if this refined perspective is adopted, the novelty of the proposed method must be **clearly differentiated** from the established line of research on **tool-use** within MLLMs, as the current setting closely resembles an MLLM leveraging a visual abstraction tool.
> >
> > ---
> >
> > **References:**
> >
> > [1] Sundaresan, Priya, et al. "Rt-sketch: Goal-conditioned imitation learning from hand-drawn sketches." 8th Annual Conference on Robot Learning. 2024.

---

> ### Author Response · Authors · 2025-11-26
> **Response to remaining concerns of weakness 1**
>
> Thanks for your reponse, we are glad that we have successfully cleared the raised questions and some of your concerns. And we also appreciate your further clarification of the remaining concerns.
>
> **W1:**
>
> 1. VAT is **intentionally not** a task-specific abstraction mechanism.
>
>   - The goal of VAT is **to reorganize the existing visual content into an object-centric, structural layout, rather than encoding task-relevant cues**. This aligns with cognitive theories of the “primal sketch,” where all objects and coarse relations are extracted first, and task relevance is resolved later during the reasoning process. In our setting, the task-specific interpretation is handled by the LLM backbone (encouraged by visual abstract). The abstraction therefore **remains general-purpose by design**.
>
>   - **While the abstraction itself is task-agnostic, different abstraction styles exist and have different properties.**
>     - Our paper (Section 5, Appendix J and M) includes discussions and comparisons of abstraction styles, and we also conducted a selection experiment where the abstraction style is routed according to the task. This is mentioned in previous rebuttal.
>     - However, these styles are **not different tools**, they are all generated from the same image without semantic enrichment or additional re-production.
>     - We hope this experiment could addresses your task-relevance concern. We demonstrate that **the representational form can be matched to the task category, even though the abstraction itself remains task-agnostic.**
>
>   - We agree that task-specific abstraction is useful. However, our goal in this work is different:
>     - VAT aims to identify a general way that improves performance across different tasks, **without injecting task-specific information.** This is why VAT **yields a consistent average improvement.**
>     - While we acknowledge that highly specialized tasks may require dedicated designs, VAT provides a generalizable reasoning benefit that holds across task types.
>
>   - VAT is therefore not designed to be an oracle for task-specific object selection; rather, it reveals that **MLLMs benefit from receiving a general purposed abstraction before reasoning.**
>
> 2. Context-aware abstraction is an exciting next step, but beyond the scope of the current work.
>
>   - Our goals in this work are:
>
>     (1) To demonstrate that MLLMs possess the inherent capability to reason using visual abstractions
>     (2) To provide a systematic analysis of this paradigm.(**which you acknowledged in your initial review, we are truly grateful for your acknowledgement!**).
>
>   - With this foundation, developing **context-aware or model-generated visual abstractions becomes a natural next step**. We fully agree that such capabilities would further enhance VAT. However, this requires a separate, more extensive investigation, and would not fit within the page constraints of this paper.
>
>   - We also noticed that some recent papers pursue a similar goal and propose insightful context-aware approaches for generating visual thoughts during the reasoning process for maze-like spatial reasoning and navigation tasks [1,2]. Although these works primarily focus on tasks like maze and frozen lake (these are grid-like navigation tasks that require spatial reasoning), they shed light for us and future researchers on a more general and applicable “thinking with images” mechanism.
>
>   - We view our contribution as **establishing a necessary foundation**, both conceptually and empirically, for future work where the abstraction becomes: model-generated, context-conditioned, and task-aware. Such directions are highly promising, and we believe they deserve a standalone paper rather than an appendix-level extension.
>
> [1] Li, C., et al. (2025). Imagine while reasoning in space: Multimodal visualization-of-thought. arXiv preprint arXiv:2501.07542.
>
> [2] Zhang, H., et al. (2025). Latent Sketchpad: Sketching Visual Thoughts to Elicit Multimodal Reasoning in MLLMs. arXiv preprint arXiv:2510.24514.

---

> ### Author Response · Authors · 2025-11-26
> **Response to remaining concerns of weakness 2**
>
> **W2:**
>
> We appreciate the reviewer’s conceptual refinement. We agree that the term should emphasize the reasoning process using the abstraction, rather than the extraction process. To address this more explicitly:
>
>
> 1. Clarifying the terminology.
>   - **We refer to “Visual Abstract Thinking” as the reasoning mode, not the extraction tool.** The thinking process is a distinct cognitive mode, shifting the model into a symbolic or structural reasoning channel rather than a pixel-perception channel.
>   - We appreciate the suggestion of "thinking with visual abstract". **It more directly captures this intended meaning of a reasoning mode.** We will adopt this in our revision.
>
> 2. If adopting “thinking with visual abstraction” terminology, **VAT is still not equivalent to tool-use.**
>
>   - Your suggested “thinking with visual abstractions” is precise. Even under this refined perspective, VAT differs from tool-use in novelty:
>     - Tool-use studies focus on **extending the model’s capability** via external modules. VAT enables an inherent mode the model can think in.
>     - Our RL results in previous rebuttal show that the model can **self-generate** and **self-utilize** this mode **without relying on any external tool.**
>
>   - Thus, the novelty lies in identifying and validating **visual abstraction as a general, reusable cognitive substrate for multimodal reasoning, rather than proposing a new external tool pipeline.**
>
> 3. VAT is fundamentally different from tool-use research.
>
> We clarify the conceptual differences emphasized by tool-use literature:
>
>  | Aspect | Tool-use MLLMs | VAT |
>  | :--- | :--- | :--- |
>  | Purpose | Obtain new, enhanced, or task-specific information | Re-encode existing information in a more reasoning-friendly format|
>  | Output | Adds knowledge not present in raw inputs | No new information; strictly information-reduced |
>  | Dependency | Performance depends on external tool’s capability | RL experiments show VAT can be internalized |
>  | Conceptual novelty | Delegation to an external process | Establishing a new reasoning pathway for MLLMs |
>
> 4. **We have modified the corresponding terminology in our revision and incorporated those valuable clarifycation.** These lines are maked in orange.

---

### Official Review · Reviewer_mANa · 2025-10-27

**Soundness:** 2
**Presentation:** 3
**Contribution:** 3
**Rating:** 4
**Confidence:** 4

**Summary:**

This paper introduces Visual Abstract Thinking (VAT), a novel multimodal reasoning paradigm. It involves feeding both the original image and an abstract visual representation (e.g., contour maps, sketches) to Multimodal Large Language Models. The aim is to guide models in focusing on critical structures and spatial relationships while reducing interference from redundant details. The study empirically demonstrates that VAT enhances MLLM performance across various visual reasoning tasks, including perception, relational reasoning, and spatial reasoning, offering advantages in computational cost and inference time compared to conventional methods.

**Strengths:**

1. The paper introduces Visual Abstract Thinking , a new approach that re-imagines how MLLMs process visual information.   By reducing runtime and token consumption compared to existing explicit thinking methods, it offers a more cost-effective approach to multimodal tasks.

2. VAT consistently achieves empirical improvements in various visual reasoning tasks, particularly those focused on object-centric perception, relational reasoning, and spatial understanding. These consistent gains across several benchmarks highlight its effectiveness within its evaluated scope.

**Weaknesses:**

1. While VAT claims to reduce cognitive load through visual abstraction, introducing a second image input increases the model’s visual payload. The paper should clarify how these abstracts genuinely reduce redundancy rather than simply adding another modality. A quantitative analysis such as mutual information between original and abstract inputs, or a correlation between abstraction fidelity and task performance would better substantiate the claim that VAT acts as a filter, not an augmenter.

2. VAT performs well on structure-sensitive tasks, but its reliance on simplified visual representations (e.g., edges, silhouettes) risks omitting critical cues such as color, texture, or fine-grained details. This raises legitimate concerns about generalizability to tasks like color counting, material identification, or emotion recognition, where such information is diagnostic. In these cases, the abstract may introduce noise rather than clarity, potentially harming performance rather than helping it.

3. The paper treats visual abstraction as a black-box input. To better understand VAT’s mechanism, it would be valuable to examine how the model internally weights or attends to the abstract versus the original image particularly in failure cases. Do attention maps focus on salient regions in the sketch? Is the abstract guiding structural reasoning, or is the effect driven largely by prompting? Such analysis would distinguish between a true shift in reasoning behavior and a superficial prompt effect.

4. The inspiration from human abstract thinking is compelling, but the link between VAT’s design and perceptual cognition remains implicit. Even a speculative discussion connecting the chosen abstractions to principles like Gestalt grouping, visual saliency, or cognitive load theory could help position VAT not merely as an engineering trick, but as a step toward cognitively plausible multimodal reasoning.

**Questions:**

1. The paper notes that VAT’s benefits are limited, and sometimes even detrimental, for smaller models (e.g., Qwen2.5-VL-7B) compared to larger ones. This suggests that VAT might not be a universally beneficial technique, but rather a "large model privilege" or a method whose efficacy scales with model capacity. Does this imply that smaller models require a different, perhaps even more drastically simplified, form of visual abstraction to avoid overburdening their limited representational capacity?

2. All experiments in the paper are conducted on static image benchmarks. Applying VAT to dynamic scenarios, such as video understanding or sequential reasoning, presents additional challenges regarding temporal consistency of abstractions. Should abstracts generated for video frames maintain coherence across time, or should they adapt dynamically?

---

> ### Author Response · Authors · 2025-11-25
> **Response to Reviewer mANa (1 of 4)**
>
> Thank you for acknowledging the effectiveness and efficiency of VAT. We sincerely appreciate your comprehensive review, and we respond to the weaknesses and questions as follow:
>
> **1. Concern of redundancy reduction via VAT (Weakness 1).**
>
> We respond to this concern via quantitative analysis of ablation study to show the correlation between abstraction fidelity and task performance and the ablation study of VAT prompting formats. We recall ablation study (In Section 4.3) here:
>
>   - **Gradually presenting visual abstraction to record its correlation with task performance**:
>
>     - As demonstrated in Figure 4 (i), we first present and convert regions without groundtruth clues, and then gradually presenting the region with clues. This allows us to record the change of task performance when the correlation of presented visual abstraction increases.
>     - Table 3 shows that **visual abstracts progressively reveal groundtruth regions, and accuracy rises accordingly**. Figure 5 (a) and (b) visualise that model confidence spikes specifically when essential cues (groudtruth in visual abstraction format) appear. **These findings collectively indicate that VAT primarily plays a filtering role rather than acting as a generic augmentation.**
>
>   - **Mixture of original image and visual abstraction to show that VAT helps reduce the influence of redundancy**:
>
>     We also conduct an ablation study that mixes the original image with visual abstract with experimental setting demonstrated in Figure 4 (ii), and report its resulting trends in Figure 5 (c). **When presenting more region as visual abstraction in a mixed image, the task performance also increases.** These demonstrate that VAT does guide models to more necessary information.
>
>   - **Ablation on VAT prompting format show that the effectiveness of VAT does not simply comes from adding another image**:
>
>     In Table 4, we compared VAT against a baseline where the raw image is simply repeated ("VAT w/ Img"). **VAT significantly outperforms this repeated-image baseline (63.09% vs. 60.41%).** This performance gap confirms that the gain stems from the abstraction mechanism removing noise, rather than the increased visual payload itself.
>
> **2. Concerns on omitting details when converting images into visual abstractions (Weakness 2).**
>
>   We appreciate this observation. We acknowledge that abstract representations reduce texture details. **In fact, this aligns exactly with our analysis in Section 5.1 that discusses these trade-offs.** Rather than introducing noise, the abstraction plays a different role to reflects a specific cognitive design:
>
>   - **VAT simulates the specific cognitive act of "structural abstraction"**:
>
>     Different tasks require different thinking modes. Just as humans deliberately use simple sketches to understand complex logic, VAT filters out "critical cues" only when they act as distractors. This mimics the biological "primal sketch" [1] and schematic thinking [3], effectively reducing cognitive load for structural reasoning.
>
>   - **The dual-input strategy helps preserve fine-grained details, instead of losing details.**
>
>     - We do not discard the original image. The visual abstract acts as a structural index, helping the model locate objects amidst clutter (coarse localization). The model then refers to the preserved original image to retrieve specific attributes like color or texture (fine identification).
>
>     - For example, we provide additional results on two fine-grained tasks, MME-Color and V*(GPT-4V-Hard), using GPT-5. The results demonstrate substantial improvements: on MME-Color, we observe gains of **+20.00**(ACC) and **+8.34** (ACC+) with VAT (rising from 73.33 to 93.33 in ACC, and 88.33 to 96.67 in ACC+) ; while VAT improves V*(GPT-4V-Hard) by **+11.77** (rising from 52.94 to 64.71). These findings indicate the effectiveness of VAT on fine-grained perception tasks.
>
>     _Note: According to the MME paper, ACC+ denotes the strict accuracy where an instance is considered correct only if both associated questions are answered correctly._
>
>   - We acknowledge that abstraction is not uniformly beneficial across all tasks, and VAT naturally performs better on tasks aligned with structural reasoning. This motivates future work on adaptive mechanisms that can dynamically balance concrete and abstract visual processing.

---

> ### Author Response · Authors · 2025-11-25
> **Response to Reviewer mANa (2 of 4)**
>
> **3. Concerns on internal mechanism and attention analysis (Weakness 3).**
>
>   Thanks for this suggestion. As proprietary models do not allow internal inspection, we managed to analyse the internal mechanism via our designd ablation study and report the following findings:
>
>   - **Our ablation study is alternative for attention analysis to distinguish reasoning from prompt effects for proprietary models.**
>
>     Since internal inspection of model weights is restricted for proprietary models such as GPT and Gemini, we used a rigorous ablation in Section 4.3. We controlled the input content, according to settings in Figure 4, to test if the model truly uses the abstract for reasoning or merely benefits from the prompt format, and how these varying visual abstractions affects the task performance.
>
>   - **Log-probability serve as a proxy signal for attention shifts, confirming genuine reasoning.**
>
>     Figure 5 in our paper demonstrates that the log probability of the correct answer rises sharply **exactly when the visual abstract of the GT region is introduced**. This alternatively reflects the attention shifting to the structural cue to resolve the query.
>
>   - **Extracting attention maps is technically infeasible due to model architecture constraints.**
>
>     Proprietary models (e.g., GPT-5, Gemini-2.5) are black boxes. For open-source models, **the Vision Transformer (ViT) typically encodes images individually**, preventing the extraction of unified attention maps that explicitly show interactions between the original image and the visual abstract during the encoding stage.
>
> **4. Cognition Theory Support (Weakness 4).**
>
>   We sincerely appreciate your recognition of the inspiration drawn from human abstract thinking. In the final version, we will include a dedicated discussion linking VAT to foundational cognitive principles:
>
>   - **VAT mimics the cognitive pre-processing of early vision.** It aligns with David Marr’s "primal sketch" theory [1]. By converting raw images into visual abstracts, VAT offloads the low-level burden of extracting edges and contours, allowing the model to focus on higher-level processing.
>
>   - **Abstraction acts as a saliency filter to prioritize structure,** aligning with Biederman’s Recognition-by-Components theory [2]. By filtering out high-frequency texture noise, VAT guides the model to focus on geometric features.
>
>   - **Structural clarity enables high-level schematic reasoning.** This mirrors Barbara Tversky’s findings on human cognitive strategies [3]. As humans use simple sketches to understand complex concepts, VAT implements this computationally to facilitate precise logical inference.
>
>   - **These impactful theories collectively verify that visual abstract is an important cognitive mechanism.** While recent works in language reasoning have begun to explore "reasoning abstractions" to guide algorithmic deduction [4], **VAT achieves this strategy to the visual domain, establishing a visual-thinking framework that parallels the efficacy of abstraction in human thought.**
>
> [1] Marr, D. (1982). Vision: A Computational Investigation into the Human Representation and Processing of Visual Information. MIT press.
>
> [2] Biederman, I. (1987). Recognition-by-components: A theory of human image understanding. Psychological Review.
>
> [3] Tversky, B. (2011). Visualizing thought. Topics in Cognitive Science.
>
> [4] Qu, Y., et al. (2025). Learning to Discover Abstractions for LLM Reasoning. ICML 2025 Workshop on Programmatic Representations for Agent Learning.

---

> ### Author Response · Authors · 2025-11-25
> **Response to Reviewer mANa (3 of 4)**
>
> **5. VAT is a "large model privilege" method (Question 1).**
>
>   We argue that the limited efficacy on smaller models stems primarily from their inherent understanding and reasoning bias rather than the complexity of the visual abstract.
>
>   - **Small models can acquire Visual Abtract Thinking capabilities via training.**
>
>     To validate that this is a training issue rather than a capacity limit, we fine-tuned the smaller **Qwen 2.5-VL-3B** model using Reinforcement Learning (RL) to incentivize the usage of visual abstracts. As shown in the table below, the RL-trained model exhibits significant performance improvements (Avg Gain +4.52%). This confirms that "Thinking with Images" is a learnable paradigm available to smaller architectures through targeted training. We will provide detailed training steps in our revision.
>
>     | Method | Odd-One-Out | Visual Illusion | Acti View | Text VQA-GT | blink-spatial | blink-counting | blink-sem_corr | blink-vis_corr | Avg. Gain |
>     | :--- | :---: | :---: | :---: | :---: | :---: | :---: | :---: | :---: | :---: |
>     | **Qwen-2.5-VL-3B** | 0.00 | 57.64 | 69.38 | 55.32 | 78.32 | 60.83 | 30.94 | 37.21 | - |
>     | **VAT-3B** | 21.57 | 61.81 | 59.69 | 69.43 | 83.92 | 61.67 | 28.78 | 38.95 | +4.52 |
>
>   - **Current small models are dominated by linguistic priors.**
>
>     Smaller VLMs, such as Qwen2.5-VL-7B, rely heavily on internal linguistic patterns [5] and struggle to integrate supplementary visual tokens. This tendency is reinforced by post-training paradigms emphasizing text-based Chain-of-Thought (CoT). Instead of "thinking with images", they revert to "thinking with language", bypassing the structural cues provided by VAT.
>
>   - **Large models demonstrate emergent "Thinking with Image" reasoning ability.**
>
>     In contrast to smaller models, larger models show a better balance between visual and textual processing. Their remarkable gains via VAT (GPT-5 +2.21 vs Qwen +0.94) suggest that utilizing visual abstractions is an emergent capability that scales with model size.
>
> **6. Performance on video tasks (Question 2)** (the rest response to this question can be found in the next part)
>
>   To investigate VAT's performance on video tasks, we conduct experiments with VSI-Bench[6]. Experimental settings, results and findings are as follows:
>
>   - Experimental setting:
>     We evaluate the two mentioned settings, coherent and adaptive. In detail: (1) VAT: visual abstract of frames are coherence across time, and (2) VAT (1/4 Frame): adaptively converting frames into visual abstract, specificall, we adopt the setting containing 1/4 frame of original video.
>
> [5] Liu, C., et al. (2025). More Thinking, Less Seeing? Assessing Amplified Hallucination in Multimodal Reasoning Models. NeurIPS 2025.
>
> [6] Yang, J., et al. (2025). Thinking in space: How multimodal large language models see, remember, and recall spaces. In Proceedings of the Computer Vision and Pattern Recognition Conference (pp. 10632-10643).

---

> ### Author Response · Authors · 2025-11-25
> **Response to Reviewer mANa (4 of 4)**
>
> **6. Performance on video tasks (Question 2)** (Continued)
>
>   - Results of evaluated VSI tasks:
>     - Tasks include：Object Counting (Obj. Count), Absolute Distance (Abs. Dist.), Relative Distance (Rel. Dist.), Relative Direction (Rel. Dir.), Route Plan, Appearance Order (Appr. Order)
>     - Metrics:
>       - MRA (Mean Relative Accuracy): offers a more reliable and discriminative measurement for calculating the similarity between numerical predictions and ground truth values[6]. This metric is employed for Obj. Count and Abs. Dist. tasks.
>       - Accuracy (ACC) is employed for the rest tasks
>     - Detailed results：
>
>   | Method                    | Frame | Obj. Count MRA | Abs. Dist. MRA | Rel. Dist. ACC | Rel. Dir. ACC | Route Plan ACC | Appr. Order ACC | Avg. Gain |
> |---------------------------|-------|-----------------|----------------|----------------|----------------|----------------|------------------|-----------|
> | Qwen2.5-VL-32B-Instruct   | 8     | 18.708          | 21.391         | 39.859         | 37.881         | 31.959         | 36.246           |           |
> | +VAT                      |       | **20.195**      | **21.547**     | 39.155         | **39.596**     | **32.474**     | 36.084           | **+0.501**|
> | +VAT (1/4 Frame)           |       | 18.513          | 20.444         | **41.268**     | 37.105         | 30.928         | **38.997**       | +0.202    |
> | Qwen2.5-VL-32B-Instruct   | 32    | 34.867          | **28.621**     | **44.225**     | 40.684         | 30.928         | 34.142           |           |
> | +VAT                      |       | **35.363**      | 27.602         | 40.282         | **42.437**     | **35.052**     | 35.761           | -0.766    |
> | +VAT (1/4 Frame)           |       | 33.788          | 27.782         | 42.254         | 41.964         | 31.959         | **38.350**       | **+0.438** |
>
>   - We summarise findings of current results from two aspects: (1) how to apply abstracted frame, and (2) the discussion of applied video length:
>
>     - Comparison of VAT-video setting (referring to aspect (1)):
>       - **The time-coherent and adaptive settings benefit different types of tasks.**
>
>         - **VAT (time coherent setting) performs well on measurement-estimation and configurational tasks.** By providing a visual abstract for every sampled frames as a new abstract frame sequence, it supplies consistent structural and geometric cues across time, helping the model interpret spatial layout, object shapes, and overall scene configuration. The abstraction also removes background redundancy and offers a clearer object silhouette and scene contour, which aids spatial reasoning and magnitude estimation.
>
>         - In contrast, **the adaptive VAT (1/4 Frame) setting performs better on spatiotemporal tasks.** Since it generates one abstracted frame for every four original frames, it captures the global temporal progression while avoiding excessive redundancy. This makes it suitable for tasks such as Appearance Order, which rely on modelling a broad temporal process rather than frame-level geometric precision.
>
>     - Comparison of the length of video frames (referring to aspect (2)):
>
>       - Increasing frame count inevitably increases input length, which **enlarges the computational cost and potentially introduce more challenging cross-reference issue among frames.** As VAT double the visual input token, 32 frames setting requires a length of 64-image input, increasing the computational cost. Moreover, the expanded input dependency might also require models to present advanced ability to figure out cross-reference among original video frames and the abstraction frames, introducing a more challenging issue. Therefore, our simulated adaptive setting, VAT (1/4 Frame), is better for long videos.
>
>       - **Coherent VAT setting is more effective for short videos, whereas adaptive VAT is preferable for longer videos.** Results show that VAT performs better on 8 frames while VAT (1/4 Frame) is better on 32 frames setting.
>
>     - To conclude, VAT can be directly applied to video inputs that maintain coherence across time for low frame rate scenarios. While it requires more dynamic design for higher frame rate scenarios.
>
>   - Adapting VAT to video is a promising yet challenging research direction. Although we observed improvements from VAT in video tasks, shortages still exist. A possible extension is to revise the positional embedding. Current time-stemp encodings often treat the VAT frames and the original video as two independent sequences. Designing a position embedding that explicitly aligns the temporal indices of the abstracted and original frames may further improve performance.
>
> [6] Yang, J., et al. (2025). Thinking in space: How multimodal large language models see, remember, and recall spaces. In Proceedings of the Computer Vision and Pattern Recognition Conference (pp. 10632-10643).

---

> ### Comment · Reviewer_mANa · 2025-11-26
>
> Thank you for the rebuttal.
>
> I still have concerns regarding the response to Weakness 1. I had already seen these experiments (Fig. 4 i + ii) during the review phase, and it was precisely because I observed that performance increases as the visual abstraction reveals more regions that I became worried that the visual abstraction might simply be adding supplementary information as another modality, rather than guiding the model to retrieve the necessary information. The current rebuttal still does not clearly address this point.
>
> In addition, I do not understand why in Table 4, under the single-image setting, feeding only the visual abstraction performs better than feeding the original image. If the visual abstraction is even more effective than the original image, doesn’t that support the interpretation that Vis.Abs is functioning as supplementary information rather than guiding the model’s information selection?
>
> Moreover, in the double-image setting, the authors also need to explain why VAT can benefit from having two identical images.
>
> Regarding the response to Weakness 3: why didn’t the authors perform attention-level analysis on Qwen2.5-VL? I don’t think there are architectural constraints that would prevent this.

---

> ### Author Response · Authors · 2025-11-27
> **Response to follow-up questions (1 of 2)**
>
> Thank you for your reply and your further clarification of remaining concerns. These are very insightful comments that help us better discuss and present our work.
>
> **Weakness 1 follow-up questions:**
>
>   - Q1:
>   >performance increases as the visual abstraction reveals more regions ...... the visual abstraction might simply be adding supplementary information as another modality, rather than guiding the model to retrieve the necessary information.
>
>     - Response:
>
>       Thank you for raising this important point. We now understand that the core concern is **whether the abstraction injects new or privileged information rather than guiding the model to retrieve the necessary information.** We clarify the following:
>
>         - **The visual abstraction contains strictly less information than the original image.**
>           It removes all texture, color, shading, and fine-grained pixel cues. The only preserved elements are object identity and coarse spatial layout, both **extracted directly from the original image without any hand-crafted features.**
>         - **No semantic enrichment is introduced.**
>           VAT does not add symbols, textual labels, or relational annotations. All content **already physically exists in the source image.**
>
>         - **The gain comes from reorganization, not addition.**
>           The abstraction reorganizes the SAME information into a format that the MLLM can parse more reliably.
>
>       Thus, VAT acts as **a representational aid that surfaces inherently available information**, not an additional modality or extra clue.
>
>       VAT CANNOT function as supplementary information because the abstraction is *strictly information-reduced*. The improvement occurs because **the model often fails to utilize relevant cues in raw pixels, and the abstraction makes those cues easier to retrieve.**
>
>   - Q2:
>   >If the visual abstraction is even more effective than the original image ...... Vis.Abs is functioning as supplementary information rather than guiding the model’s information selection.
>
>     - Response:
>
>       We appreciate this question, as it emerges naturally from Table 4. The key issue is that **Table 4 averages across heterogeneous tasks**, so the conclusion depends on examining task-level behaviors.
>
>         - Table 4 is **an ablation about VAT’s prompt template, not about task-level superiority of abstraction versus raw image.**
>         - It studies **which components of the prompt contribute to VAT**. It does NOT claim the abstraction is universally superior to the original image.
>         - The task-level breakdown (Table 10 in Appendix G) shows:
>           - Tasks where Img-only **\<** Vis.Abs-only :
>
>             e.g., CoSpace-Rot-dif (50.50 --> 95.00) and Odd-One-Out (25.49 --> 31.37). These tasks rely primarily on object identity and coarse spatial structure. **These are exactly what abstraction preserves.**
>
>           - Tasks where Img-only **\>** Vis.Abs-only:
>
>             e.g., BLINK-Sem.Corr (53.96 --> 44.60) and BLINK-Spatial (82.52 --> 81.82) (the focus of BLINK-Spatial is not the overall spatial layout, but finer "left/right" concepts), which require finer-grained cues that **abstraction alone cannot supply.**
>
>       Thus, Vis.Abs-only performs well on tasks where the abstraction preserves the required information while removing distracting details from the raw image. In those cases the **abstraction helps the model find and use the existing relevant cues more reliably.** Conversely, when fine details matter, the abstraction predictably underperforms the Img-only setting.
>
>       This supports the interpretation that **VAT helps the model select and utilize existing information, not that it adds new signals.**

---

> ### Author Response · Authors · 2025-11-27
> **Response to follow-up questions (2 of 2)**
>
> - Q3:
>   >Why VAT can benefit from having two identical images?
>
>     - Response:
>
>       We clarify that this phenomenon is **not unique to VAT**. Similar performance gains from duplicated inputs have been repeatedly observed across LLMs and MLLMs. While its theoretical underpinnings remain an open problem, several empirical findings support the idea that repeated inputs can improve model performance.
>
>         - For LLMs, repeating the same input improves reasoning. For example:
>           - Xu et al. propose re-reading strategy (Re2), and show that by processing the question twice (re-reading, seeing identical qestions), reasoning performance of LLMs can be consistently enhanced across diverse benchmarks. Their explanation is that the second pass allows the model to reconstruct a more consistent internal representation, and the model effectively gains a bidirectional view of the input.
>
>           - Arora et al. [2] also found repeating the input multiple times helpful.
>
>         - For MLLMs, Zou et al. [3] propose look-twice strategy and find it helps mitigate hallucination, and consequently improves task performance.
>
>       Therefore, we hypothesis that having two identical images improves the results for similar reasons.
>
> **Weakness 3 follow-up questions:**
>
>   The attention visualization cases from Qwen2.5-VL-32B are updated in Appendix N in our revision. These help visualize the attention shift and also how visual abstract reorganizes existing visual information without adding extract information.
>
>
> [1] Xu, X., et al. (2024). Re-Reading Improves Reasoning in Large Language Models. EMNLP 2024.
>
> [2] Arora, S., et al. (2024). Just read twice: closing the recall gap for recurrent language models. In Workshop on Efficient Systems for Foundation Models II@ ICML2024.
>
> [3] Zou, X., et al. (2025). Look Twice Before You Answer: Memory-Space Visual Retracing for Hallucination Mitigation in Multimodal Large Language Models. In Forty-second International Conference on Machine Learning.

---

### Author Response · Authors · 2025-11-25
**General response**

We thank all reviewers for their time and comments. Our VAT is inspired by human abstract cognition, and we managed to confirm that MLLMs could also present similar ability and their task performance could be improved via VAT.

VAT is recognized by reviewers as **a novel reasoning paradigm (mANa, hRp8)**, delivering **superior reasoning accuracy while being efficient in costs (mANa, mHuw)**, **simple yet effective (8NEK, mHuw)**. We are glad that reviewer **8NEK, mHuw, and hRp8 find our ablation study helpful and convincing**.

We also appreciate the concerns raised by the reviewers, which help us improve and refine the paper. We will address these concerns in detail, including common issues of:
  - clarifying comparisons and the mechanism revealed by our ablation studies and analysis, (reviewer mANa, 8NEK, hRp8)
  - adding additional experiments to further demonstrate the effectiveness and generality of VAT across tasks, (reviewer mANa, 8NEK, mHuw)
  - expanding our discussion of VAT’s relation to tool-use and its potential as an internalizable reasoning paradigm. (reviewer 8NEK, mHuw)

Point-by-point responses to each reviewer can be found in the individual rebuttal sections.

---

### Author Response · Authors · 2025-12-03
**Summary of Discussion During Rebuttal Period**

## Core Contributions of Visual Abstract Thinking

1. **Novel ''Think with Visual Abstract'' reasoning paradigm** (mANa, hRp8):
We introduce VAT, a reasoning paradigm aligned with human visual cognition. VAT enables models to filter out redundant visual information while preserving higher level semantics and essential structural cues.

2. **Simple yet effective，with lower time and computational cost** (mANa, mHuw, 8NEK):
By aligning with human cognitive processing, VAT enables more accurate inference, and naturally
reduces the visual thinking burden of the model from the output side, permitting faster reasoning with significantly lower time and cost overhead.

3. **In-depth and convincing analysis on internal mechanism of VAT**  (8NEK, mHuw, hRp8):
We designed comprehensive experiments and derived analysis to demonstrate VAT's underlying mechanism and also its flexibility.

## Concerns raised by reviewers and how we address them

We also appreciate the concerns raised by the reviewers, which help us improve and refine the paper, including common issues of:

1. clarifying comparisons and the mechanism revealed by our ablation studies and analysis, (reviewer mANa, 8NEK, hRp8)

2. adding additional experiments to further demonstrate the effectiveness and generality of VAT across tasks, (reviewer mANa, 8NEK, mHuw)

3. expanding our discussion of VAT’s relation to tool-use and its potential as an internalizable reasoning paradigm. (reviewer 8NEK, mHuw)

We addressed all of concerns below with additional experiment results or detail clarifications below.


## Major Updates of Experimental Results

1. **VAT is generalizable across model scales and training/inference stage** (see `Appendix O`)
  - Further training with VAT via RL validates the generalizability of our VAT paradigm on smaller models and also the internalizability of our proposed reasonign paradigm, which successfully addressed Q1 (concern of large-model privilege) from mANa; W1 (concern of context-awareness and task-relevance) from 8NEK; and W1 & Q1 (distinction between VAT and tool use) from mHuw. **Our experiment shows VAT not only work for large proprietary models, it can also emerge for smaller models via training.**

2. **Attention Map of VAT** (see `Appendix N`)

  - Additional attention visualizations on Qwen-2.5-VL-32B show that VAT guides model attention toward essential visual elements, concepts, and structural features by suppressing redundant information (addressing W3: concerns on internal mechanism and attention analysis, from mANa).

3. **Ablations of different sketch styles** (see `Appendix M`)

  - We provided experimental results comparing different styles of visual abstracts across different tasks (addressing W3: ablations of different sketches and SoM baselines from 8NEK). **Different styles supply different abstract features, but they all present general improvements.**

4. **Investigation of temporal dynamic tasks**
  - We apply VAT on temporal dynamic videos, and discuss experimental results with both time-coherent and adaptive settings (addressing Q2 on video performance and temporal coherence from mANa). Beyond static images, **VAT is also effective for dynamic inputs, and can be applied with flexible patterns.**

5. **Prompt strictness study**
  - We analyze the influence of instruction strictness: Strict (“you must use visual abstract”), Flexible (“you may use visual abstract”), and No prompt.  (addressing Q2 on flexible prompting from hRp8.) **Results show that performance generally increases with stricter prompts**.


## Further Clarification of Theoretical and Analytical Supports

1. **Cognitive theoretic support** of Visual Abstract Thinking is added to show VAT presents the paradigm aligning with perspective of human perception and reasoning (addressing W4 from mANa, VAT’s design and perceptual cognition remains implicit).


2. **VAT captures universal structural and geometric information rather than task-specific heuristics.**
Multimodal reasoning inherently benefits from decomposing complex scenes. VAT mimics the cognitive primal sketch [1] to filter low-level noise and provide a cleaner structural layout.

3. **The dual-input setting (visual abstract + original image) preserves fine-grained details.**
The visual abstract supplies a spatial and structural prior that guides the model toward salient regions, while the original fine-grained cues, such as color, texture, and local patterns, remain fully accessible.
(addressing W2: VAT as external-tool use and lack of context awareness from mANa; and W2: concerns about VAT’s generalizability to fine-grained tasks from mHuw)


[1] Marr, D. (1982). Vision: A Computational Investigation into the Human Representation and Processing of Visual Information. MIT press.

---

### Meta-Review · Area_Chair_AABT · 2026-01-08

**Summary:**

- Mechanism & Novelty: Multiple reviewers questioned whether Visual Abstract Thinking was a novel reasoning paradigm or merely an effective external tool-use/prompting technique.

- Generality & Limitations: Reviewers expressed concerns about VAT's applicability to fine-grained tasks, its performance on smaller models, and its extension to dynamic video inputs.

- Analysis Depth: Reviewers sought deeper analysis, including internal attention mechanisms, failure cases, ablations on sketch styles (8NEK), and stronger baseline comparisons.

- Theoretical Grounding: Reviewer felt the link to human cognitive theory was implicit and needed strengthening.

**Reviewer Concerns:**

Addressed Concerns:

- Internal Mechanism： The authors provided attention visualizations for Qwen-2.5-VL, directly addressing the request. They also cogently explained the limitations for proprietary models.

- Ablation Studies: The rebuttal added extensive new experiments: comparisons of different sketch styles, a direct comparison to Set-of-Mark prompting (where VAT outperformed SoM), and a prompt strictness study.

- Performance on Fine-Grained Tasks: New results on MME-Color and V* showed clear improvements

- Video Task Performance: The authors conducted new experiments on video tasks.

- Clarification of Results and Terms: The authors corrected numerical errors and clarified the "Rand-GT" condition.

- Flexible Prompting: A new experiment comparing strict, flexible, and no-instruction prompts was conducted, providing valuable insights.

Unaddressed Concerns:

- Core Novelty vs. Tool-Use: This remains the most significant philosophical debate.

- Reduction vs. Augmentation: The author's explanation that the abstract is "strictly information-reduced" and acts as a "representational aid" is logical. However, the reviewer's puzzlement over cases where the abstract alone outperforms the original image hints at a desire for an even more fundamental information-theoretic analysis, which may be beyond the paper's scope.

- Theoretical Grounding: The authors promised to add a dedicated discussion linking VAT to Marr's primal sketch, Biederman's Recognition-by-Components, and Tversky's schematic reasoning.

**Reviewer Scores:**

Reviewer hRp8 may raise the score, and other Reviewers probably won't change the scores.

---

### Decision · Program_Chairs · 2026-01-26

Reject